TOPICAL REVIEW

# Structure mirroring function: What's the 'matter' with the funny current?

Andrea Saponaro[1] (ID) and Dario DiFrancesco[2] (ID)

[1] *Department of Pharmacological and Biomolecular sciences, University of Milano, Milan, Italy*
[2] *Department of Biosciences, The PaceLab, University of Milano, Milan, Italy*

Handling Editors: Laura Bennet & T Alexander Quinn

The peer review history is available in the Supporting Information section of this article (https://doi.org/10.1113/JP287209#support-information-section).

The Journal of **Physiology**

**Abstract figure legend** The 'funny' ($I_f$) current of cardiac pacemaker cells has been first identified in the late 1970s as a major mechanism in the generation and control of cardiac pacemaking. Decades of studies have since described the properties of the funny current and of its molecular components, HCN channels, in the heart and brain, providing the basis for interpretation of the functional role of f/HCN channels in a variety of physiological processes and clinically-relevant applications. The recent development of techniques providing unprecedented resolution of molecular details of channel proteins has allowed us to peek into the structural mechanisms underlying f/HCN channel behaviour. An obvious question arises that can now be addressed: do structural data explain the original, basic properties? Does structure 'mirror' function? Here, we show that structural data, synergistically supported by molecular dynamics simulations, confirm most of the early original results and their interpretation in terms of functional modelling. Although early functional studies of native funny channels set the stage for a global understanding of the biophysical properties of this channel family, the era of cryogenic electron microscopy can now provide resolution-revolution structural information, thus paving the way for an atomistic description of the structure/function correlates of these proteins. This comparison highlights how merging functional and structural studies can help develop a more comprehensive understanding of protein function.

**Abstract** First described in native cardiac pacemaker cells, the 'funny' ($I_f$) current provided a novel mechanism able to underlie rhythmic activity and autonomic control of heart rate. Increasing the impact of this finding, the new mechanism replaced a previous pacemaking model based on a 'fake' $K^+$ current ($I_{K2}$), shown in fact to be a 'camouflaged' $I_f$; also, a similar current in neurons ($I_h$) was found to regulate neuronal excitability. $I_f$, the first described inward current activated on hyperpolarization, had several other peculiar features, when investigated in sinoatrial node tissue and isolated cells. It had a mixed $Na^+/K^+$ permeability, had the lowest patch clamp recorded single-channel conductance, and was dually activated by voltage and intracellular cyclic nucleotides. $I_f$ activation by internal cAMP, a property key to autonomic modulation of heart rate, was shown to involve direct cAMP binding to channels. Finally, an $I_f$ blocking drug, ivabradine, was found to be suitable for the pharmacological control of heart rate in therapies against angina and heart failure. Later cloning of HCN channels, comprising the subunit components of funny channels, allowed molecular insight into the properties of $I_f$, carried by HCN4. Recently, cryogenic electron microscopy has resolved details of the HCN4 structure with unprecedented precision, providing a way to validate or refute, on a structural basis, original interpretation/modelling of experimental data. This review aims to compare elementary functional properties of $I_f$ *vs.* HCN4 protein structure. Does structure 'mirror' function? We show that the peculiar $I_f$ characteristics originally described are elegantly explained and 'mirrored' by structural features of the channel protein.

(Received 6 November 2024; accepted after revision 3 February 2025; first published online 27 February 2025)

**Corresponding author** D. DiFrancesco: University of Milano, Department of Biosciences, The PaceLab via Celoria 26, 20133 Milan, Italy. Email: dario.difrancesco@unimi.it

## Introduction

It is well established that progress in the understanding of how physiological processes occur is greatly enhanced by technical developments increasing the resolution of microscopic investigation of cellular/molecular events. The field of ion channels is no exception. Typically, recent advancements in the field of protein science, such as the cryogenic electron microscopy (cryo-EM) technique, have helped resolve, with an unprecedented precision, the structure of several ion channels.

An improved understanding of ion channel structure is not only important *per se* as a newly acquired knowledge. More significantly, deeper insight into details of the structural changes involved in channel activity allows the provision of a molecular interpretation of the channel basic properties, as described in early electrophysiological experiments often several decades before these recent developments. These include properties such as voltage-dependent kinetics, control of channel activity by internal/external modulating factors, ionic permeability, channel block by blocking agents and several more aspects of the channel performance.

This new approach thus represents a way to provide a molecular validation (or refutation) of how original experimental data first describing ion channel functional properties were interpreted and/or modelled.

HCN channels belong to the superfamily of the pore-loop cation channels, characterized by selectivity filters belonging to loops extending into an aqueous pore from one side of the membrane, which include Kv and CNG channels (MacKinnon, 1995). In mammals, they

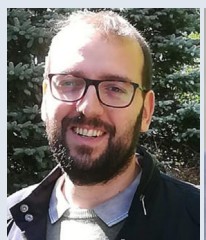
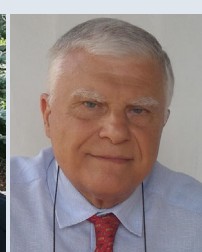

**Andrea Saponaro** is a tenure track Assistant Professor of Physiology at the Department of Pharmacological and Biomolecular Sciences, University of Milano. His research is focussed on the molecular mechanisms of ion channel gating. He has a broad background in ion channel biophysics, with specific training and expertise in structural biology, recently including high-resolution cryogenic electron microscopy and macromolecular interactions. **Dario DiFrancesco** is Emeritus Professor of Physiology at the University of Milano. In 1979, he and his coworkers first described the cardiac pacemaker ('funny') current, proposing a novel mechanism explaining generation of spontaneous activity and rate control. This initiated a new field of research in heart and brain, where HCN channels, molecular f-channel components, are today known to play fundamental roles in health and disease. HCN studies have led to clinically-relevant applications including development of a heart rate-reducing drug marketed against heart failure and angina, as well as the identification of disease-linked HCN variants.

are expressed in cardiac and smooth muscle cells, as well as in neurons, where they serve many important physiological functions, including cardiac pacemaking and neuronal excitability. HCN channel expression has also been reported in a variety of non-excitable cells in many different systems and organs, where their functional relevance remains to be clarified (Benzoni et al., 2021).

Similar to Kv and CNG channels, HCN channels are tetramers, but they have some distinctive properties. For example, Kv channels are gated by voltage, and CNG channels are gated by cyclic nucleotides, whereas HCN channels are *dually* gated by both. Also, curiously, HCN channels open on *hyperpolarization*, not on *depolarization* as for Kv channels, despite structural similarities.

An exception to the rule of depolarization-induced opening for Kv has so far only been found in KAT1 from *Arabidopsis thaliana* (Latorre et al., 2003) and the archaebacterial MVP (Sesti et al., 2003).

The atypical features of HCN channels had in fact become apparent before their cloning, with the identification of a native current dubbed 'funny' ($I_f$ current) because of its unusual properties, functionally described first in cardiac pacemaker cells and then in neurons ($I_h$ current). Searching for the 'funny' channel gene(s) eventually led to the cloning of the HCN channel family in the late 1990s (Gauss et al., 1998; Ludwig et al., 1999; Santoro et al., 1998).

The basic properties of the funny current were described in early electrophysiological experiments, first in multicellular preparations and then in single cells (DiFrancesco, 1993; DiFrancesco & Tortora, 1991). Early experimental findings on functional properties of native channels were then confirmed in HCN channels after their cloning (Accili et al., 2002; Baruscotti & DiFrancesco, 2004; Robinson & Siegelbaum, 2003; Wahl-Schott & Biel, 2009). The aim of this review is to compare elementary functional properties of the funny current and the HCN protein structure. Does structure 'mirror' function? As we discuss below, structural details of the HCN channel proteins elegantly explain and faithfully 'mirror' the functional properties of the funny current as described in the original experiments.

### Early description of $I_f$ properties

**$I_f$ is an inward current activated on hyperpolarization.** The elementary properties of the funny current were first described by electrophysiological investigation of spontaneously beating whole tissue preparations from the rabbit sinoatrial node (SAN) (Fig. 1) (Brown et al., 1979) and soon after the re-interpretation of the $I_{K2}$ current (Fig. 2), also from Purkinje fibres. When, in the mid 1980s, techniques for single-cell isolation became available, its

properties were confirmed in single pacemaker cells (DiFrancesco et al., 1986).

V clamp analysis showed that the current is *inward* and activates on *hyperpolarization* at voltages in the range where the slow diastolic (pacemaker) depolarization (DD) phase of the action potential develops (Fig. 1*A*). Because an inward current has a depolarizing effect, these properties were exactly those expected for a mechanism contributing directly to the generation of the DD. Involvement in the generation of the DD was also evident from another, physiologically relevant property: adding adrenaline to the perfusing solution increased the inward $I_f$ (Fig. 1*A* and *B*), accounting for an increased steepness of the DD, and consequent acceleration of the spontaneous rate of pacemaker cells. As well as contributing to *initiating* of pacemaker activity, the novel $I_f$-based mechanism was thus involved in controlling pacemaker frequency and contributing to mediate sympathetic rate modulation.

Here, however, we want to address more specifically the biophysical aspects of the funny current. The development of enzymatic isolation methods in the mid 1980s allowed the investigation of $I_f$ in single SAN pacemaker cells (DiFrancesco et al., 1986). The fact that isolated SAN cells beat spontaneously and express, at physiological voltages, a large $I_f$ otherwise not found in working muscle, quiescent cardiac cells, by itself provides evidence that membrane expression of funny channels and pacemaker activity are strictly correlated.

The most elementary $I_f$ properties measured with specific patch clamp protocols in single pacemaker cells are shown in Fig. 1*C*. The current is activated on hyperpolarization, with an activation voltage approximately spanning the interval –50 to –80 mV, and has a fully-activated $I/V$ relationship reversing at voltages close to –10 mV.

**$I_f$ is carried by both Na$^+$ and K$^+$.** Following its first description, early studies of the $I_f$ current were devoted to the investigation of its ionic permeability. The pacemaker current in Purkinje fibres had been previously interpreted as a pure K$^+$ current, outward and deactivating on hyperpolarization ($I_{K2}$ model), based on the erroneous interpretation of its time course on hyperpolarization. This was shown to be the misleading result of the superimposition of two events: the activation of $I_f$ and the decay of the inward K$^+$-rectifying current generated by depletion of extracellular K$^+$ ions from extracellular clefts during strong hyperpolarizing steps (DiFrancesco, 1981a). After its 'reinterpretation,' from $I_{K2}$ to $I_f$, measurement of $I/V$ curves in Purkinje fibres showed that the $I_f$ reversal potential depends on both external Na$^+$ and K$^+$ concentrations, indicating that the current has a mixed Na$^+$-K$^+$ permeability (Fig. 2) (DiFrancesco, 1981b; DiFrancesco, 1982).

These experiments highlighted an odd property of the current: increasing the external $K^+$ concentration increased the $I_f$ conductance (Fig. 2*B*), whereas no such effect was seen upon changing external $Na^+$ (Fig. 2*A*). $I_f$ was totally abolished at zero external $K^+$ (Fig. 7 of DiFrancesco, 1982) but, curiously, only in the inward direction, whereas a substantial current was recorded in the outward direction. $K^+$-dependent activation of the inward current could be interpreted in terms of a Michaelis–Menten process with a one-to-one $K^+$ ion /channel stoichiometry and a dissociation constant of $K^+$ binding of ~44 mM (DiFrancesco, 1982).

Investigating, in the same study, the action of different external cations revealed that $I_f$ is blocked by mono-valent cations such as $Cs^+$ and $Rb^+$. Block by $Cs^+$, for example, of the *inward* $I_f$ was strongly concentration- and voltage-dependent, increasing at more negative potentials, and could be described by a block model where one $Cs^+$ ion blocks one channel by entering the channel from outside and binding within the pore after crossing a given fraction (0.71) of the transmembrane voltage drop. The action of $Cs^+$ ions on the *outward* component, however, was completely different because, similar to $K^+$, $Cs^+$ increased the outward current. Data obtained with variable external concentrations of $K^+$, $Na^+$ and $Cs^+$ could be explained by assuming that '*the pacemaker channel possesses an activating site to which $Cs^+$, $K^+$, and $Na^+$ bind competitively, and that the binding of $K^+$ and $Cs^+$*

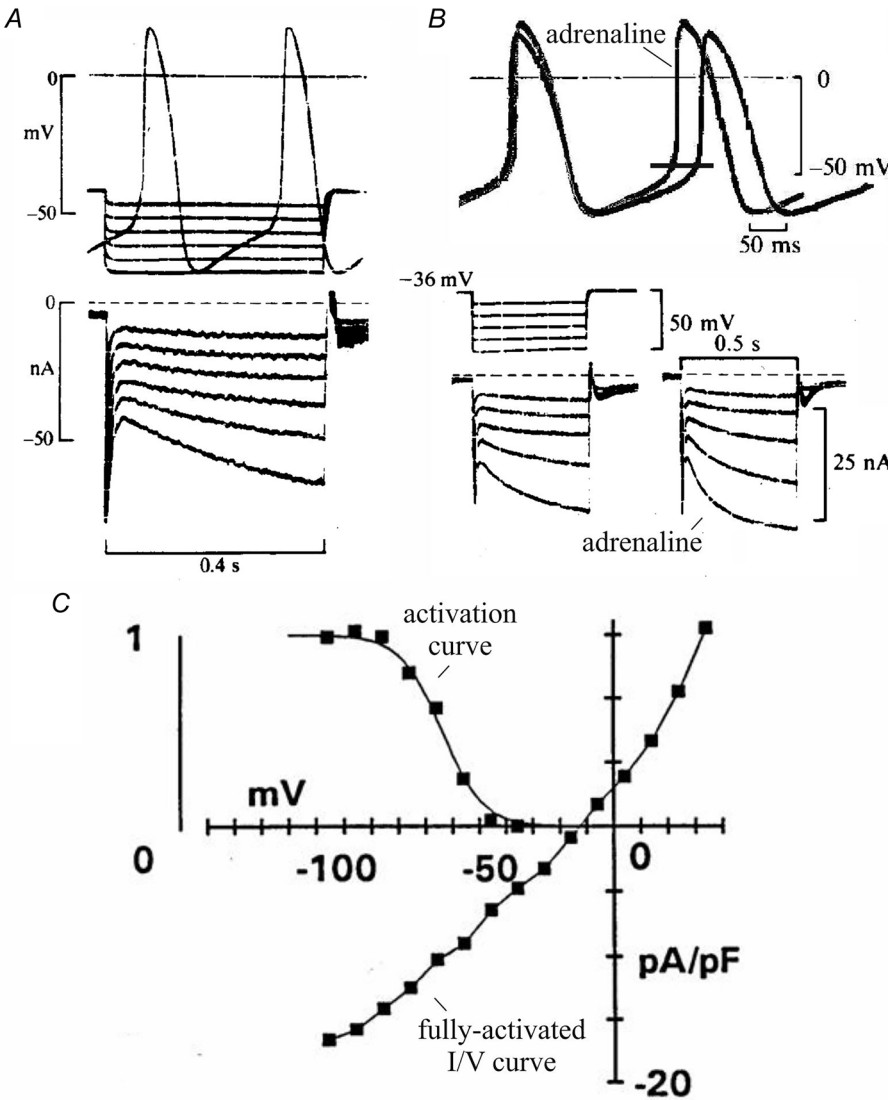

**Figure 1. Basic properties of the funny current**
*A*, $I_f$ is an inward current activated on hyperpolarization at voltages within the diastolic depolarization range. *B*, adrenaline increases $I_f$, thus generating a steeper DD phase and rate acceleration; reproduced with permission from Brown et al. (1979) (modified). *C*, basic properties are described by the activation curve and the fully-activated *I/V* relationship.

*triggers the channel activating process*' (DiFrancesco 1982). Only recently, as we discuss below, have these results found a molecular interpretation based on structural data.

**Single f-channels have a very small unitary size.** The first single f-channel recording (DiFrancesco, 1986) (Fig. 3) was a demanding achievement because, as it happened, the size of $I_f$ unitary currents turned out to be among the smallest ever recorded with the patch clamp technique. To overcome the difficulty of resolving these extremely low amplitude records (tens of femto-ampere), a modified patch clamp protocol was devised where two pipettes, one in the whole-cell and the other in cell-attached configuration, were sealed on the same

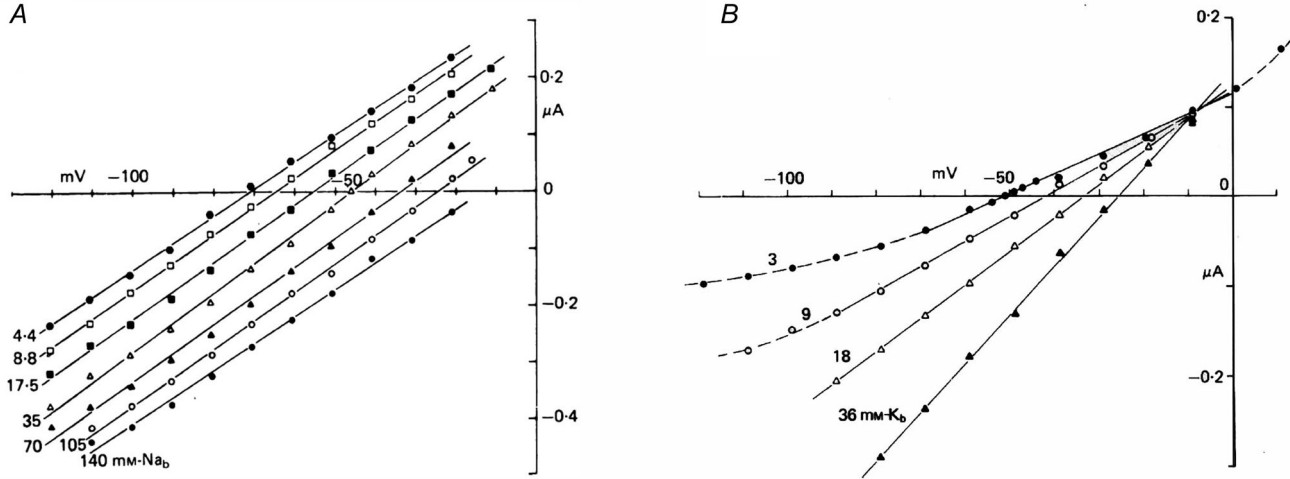

**Figure 2. Mixed Na⁺-K⁺ permeability of f-channels**
Fully-activated *I/V* relationships of $I_f$ measured in Purkinje fibres in different external Na⁺ (*A*) and K⁺ concentrations (*B*). Note that the $I_f$ conductance is not dependent on external Na⁺, but increases at higher external K⁺ (reproduced with permission from DiFrancesco, 1981b).

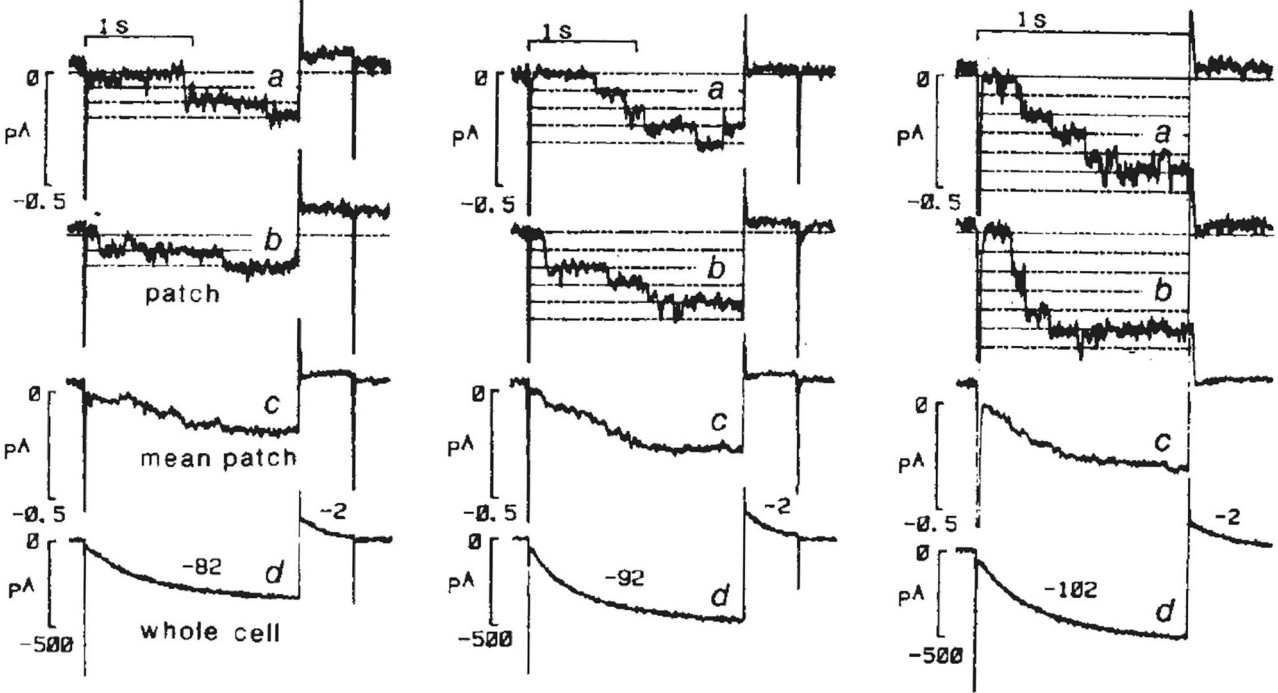

**Figure 3. Single f-channel**
Single f-channel recordings from SAN pacemaker cells with a two-pipette configuration, allowing simultaneous recording of whole-cell and single channel traces. The single-channel conductance is 0.98 pS (reproduced with permission from DiFrancesco, 1986).

cell. The arrangement had two advantages: first, because V-steps were applied from the whole-cell pipette, the patch clamp pipette recorded single-channel events without overlapping, much larger capacitive transients, yielding a much improved signal resolution; second, whole-cell and single-channel traces were recorded *simultaneously* from the same cell, which allowed comparison of the ensemble average of single-channel traces with the whole-cell trace recorded simultaneously. As apparent from Fig. 3, the similarity between time courses of ensemble average and whole-cell traces at all voltages is striking, confirming that the cell-attached records represent single f-channel events. From these data, a single f-channel conductance of 0.98 pS was calculated. This low value was later confirmed using cell-free measurements (DiFrancesco & Mangoni, 1994).

**Autonomic transmitter modulation of the funny current is mediated by voltage shifts of the activation curve.** When first described, the funny current was shown to be increased by adrenaline (Brown et al., 1979). Later studies showed that the current is also modulated by the parasympathetic autonomic transmitter, ACh, but in the opposite direction (DiFrancesco & Tromba, 1987, 1988a; DiFrancesco et al., 1986). The two autonomic transmitters modify $I_f$ by shifting its activation curve along the voltage axis, as shown in Fig. 4*A*, with no modification of the fully-activated *I/V* curve. Catecholamines shift it to the right, thus increasing $I_f$ availability during the DD, whereas ACh shifts it to the left, thus decreasing $I_f$ availability (Accili et al., 1997; Callewaert et al., 1984; DiFrancesco & Tromba, 1987, 1988b; DiFrancesco et al., 1986).

This evidence provided a mechanism contributing to one of the fundamental cardiovascular functions (i.e. modulation of heart rate by the autonomic nervous system) and had an impact in cardiac physiology *per se*. However, to identify the process controlling the voltage dependence of the $I_f$ activation curve, more detailed cellular investigation was required.

**Direct binding of intracellular cAMP gates funny channels.** Specific studies aiming to investigate autonomic $I_f$ modulation (DiFrancesco & Tromba, 1987, 1988a, 1988b) revealed that the voltage dependence of the $I_f$ activation curve is controlled by a G protein-dependent pathway involving intracellular cAMP, which acts as a second messenger mediating regulation of cardiac rate by the autonomic nervous system.

To understand how cAMP exerts its shifting action on the f-channel activation curve, experiments were devised using large-tipped pipettes, with the aim of obtaining macro-patches containing hundreds/thousands of f-channels (Fig. 4*B*) (DiFrancesco & Mangoni, 1994; DiFrancesco & Tortora, 1991). This avoided the difficulty of recording extremely small single-channel currents, at the same time as allowing fast perfusion of the intra-cellular side of the membrane. The evidence collected from this configuration confirmed that, as expected, cAMP acts as a second messenger in the $I_f$ channel control. More surprisingly, cAMP acted by direct binding to the cytoplasmic side of the channel, rather than, as could have been expected, by a protein kinase A-mediated mechanism such as the one controlling L-type $Ca^{2+}$ channels. This feature attracted the interest of researchers and remains one of the most intensely investigated aspects of f-channels. Although not immediately recognized, it also represented the first evidence that f-channels share properties with an apparently distant family of channels, the CNG channels, which are, however, voltage independent. Cloning of HCN channels (see below) and cryo-EM reconstruction of their structures has helped resolve, as we discuss below, some of the basic mechanisms controlling cAMP-dependent channel gating.

A hint to a possible mechanism explaining how cAMP binding leads to a positive shift of the activation curve

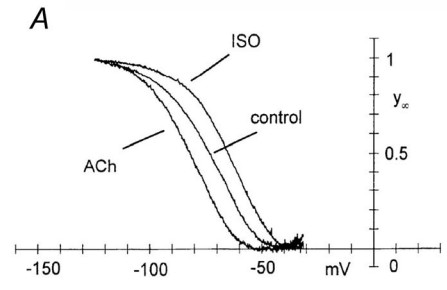
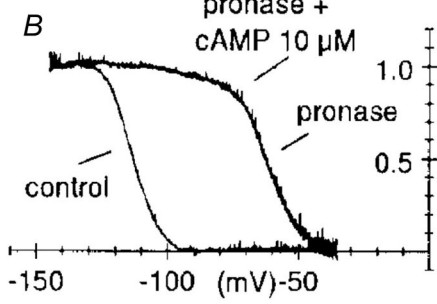

**Figure 4. f-channel modulation by cAMP**
*A*, data from whole-cell voltage ramps applied to a single SAN myocyte, showing the shifts of the $I_f$ activation curve caused by isoprenaline (right) and ACh (left) (from Accili et al., 1997). Later studies showed that the position of the activation curve is controlled by intracellular cAMP (DiFrancesco & Mangoni, 1994; DiFrancesco & Tortora, 1991). *B*, exposure of a macro-patch to pronase causes a giant shift of >+56 mV. Further addition of cAMP has no effect (reproduced with permission from Barbuti et al., 1999).

was gained by exposing inside-out macro-patches to pronase, a non-specific protease (Barbuti et al., 1999). Exposure to pronase caused a large positive shift of the $I_f$ activation curve (over 56 mV), following which further addition of cAMP was ineffective (Fig. 4*B*). The data could be interpreted to indicate that: (1) the intracellular portion of the channel (more specifically the C-terminus) exerts a constitutive inhibitory action on channel opening, and its truncation fully abolishes the inhibition; (2) cAMP binding to the channel acts by partially removing inhibition; and (3) when the inhibitory effect is fully removed by pronase, cAMP has no further action.

Knowledge of this basic structural mechanism set the background for further investigation of the processes underlying partial removal of the constitutive inhibitory action of the C-terminus by cAMP (Chen, Wang et al., 2001; Viscomi et al., 2001; Wainger et al., 2001; Wang et al., 2001; Zagotta et al., 2003).

**Selective blockers of funny channels as heart rate-reducing agents.** The involvement of f-channels in the generation of spontaneous activity and cardiac rate control makes them obvious pharmacological targets.

Essentially the only phase of the action potential affected by $I_f$ changes is the diastolic depolarization of pacing cells, and drugs selectively blocking funny channels are therefore expected to slow rate specifically, with no or little side-effects. Rate lowering is important in ischaemic heart disease, angina and heart failure, and drug companies have therefore pursued the search for specific f-channel blockers for clinical use subsequent to its first description. Of the several f-blocking compounds produced, ivabradine, marketed as the first selective and specific f-channel blocker and a pure heart rate-reducing agent, is the only one commercially available at present (DiFrancesco & Camm, 2004). A Modernized Classification of Cardiac Antiarrhythmic Drugs (Lei et al., 2018) has recently proposed that $I_f$ inhibitors such as ivabradine represent a new antiarrhythmic class (class 0), and identified the $I_f$ reduction as a target for antiarrhythmic treatment.

Ivabradine, perfused extracellularly, acts by partial block of $I_f$ (Fig. 5*A*), leading to a slowing of diastolic depolarization and pacemaking rate (Fig. 5*B*). Potential clinical use of the drug fuelled increased interest for a detailed investigation of its blocking action.

Among the block properties, an atypical one found in native pacemaker cells (Bucchi et al., 2002) was the current-dependence of block. For example, block was removed by large hyperpolarizing steps, but not because of an intrinsic *voltage*-dependency, because the same protocol failed to remove block when the inward current flow was abolished by Cs$^+$, indicating *current*-dependency. This and other properties were explained assuming that for block to occur, ivabradine needs to enter the water-filled cavity below the pore, where it competes with permeating ions at a binding site within the permeation pathway (Bucchi et al., 2002).

## Structure mirroring function

**HCN channels are the alpha subunits of native funny channels.** The data discussed above originate from experimentation on native funny channels and provide the basic features of their functional properties. However, it was only with the cloning of funny channels in the late 1990s that a molecular view of these features became accessible (Gauss et al., 1998; Ludwig et al., 1999; Santoro et al., 1998).

In humans, four genes (*hcn1-4*) encode for the Hyperpolarization-activated Cyclic Nucleotide-gated (HCN1-4) proteins, respectively, identified as the molecular determinants of the pore-forming subunits of funny channels. Although HCN1 and HCN2 are predominantly expressed in the nervous system, both central and peripheral, no obvious organ specific expression was assigned for HCN3 (Robinson & Siegelbaum, 2003). HCN4 was instead recognized as the predominant isotype in the SAN pacemaker cells (Altomare et al., 2003; Brioschi et al., 2009; Liu et al., 2007). Note, however, that several studies of animal models indicate that other isotypes, typically HCN1 but also HCN2, are expressed in SAN cells (Marionneau et al., 2005; Moroni et al., 2001; Zicha et al., 2005), indicating that heteromeric channel constructs may contribute significantly to determine the properties of the native SAN funny channels.

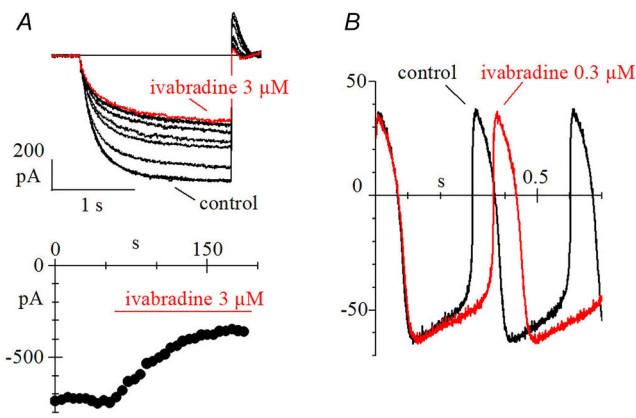

**Figure 5. Action of ivabradine**
The drug ivabradine partially blocks $I_f$, (*A*) thus slowing the diastolic depolarization and spontaneous rate (*B*) (reproduced with permission from DiFrancesco, 2010).

**Resolution-revolution of protein structures: HCN channels.** As mentioned above, cloning of HCN channels has provided a first basis for a molecular approach to their properties. Substantial progress in the exploration of protein structure was then achieved with the development of X-ray crystallography and cryo-EM, techniques that strongly amplified the resolution of structural studies. Crystal and NMR structures of the isolated C-terminal cyclic nucleotide-binding domain (CNBD) of HCN1, 2 and 4, in synergy with biochemical/biophysical experiments, described conformational changes associated with cAMP binding, thus clarifying the starting point of the cAMP pathway in HCN channels (Akimoto et al., 2014; Lolicato et al., 2011; Saponaro et al., 2014, 2018; Xu et al., 2010; Zagotta et al., 2003). Although these partial structures undoubtedly provided useful information, in the absence of full-length protein structures, the isolated C-terminal domains alone were not sufficient to fully describe the atomic details of the cAMP regulation of HCN channels, and, more importantly, the differences in the extent of cAMP modulation between isotypes. Several studies later showed that this isotype specificity lies mainly in the cytosolic region, and its interaction with the transmembrane domain of the channel (Alvarez-Baron et al., 2018; Chen et al., 2007; Kusch et al., 2010, 2012; Lolicato et al., 2014; Wang et al., 2001).

More recently, the cryo-EM technique has allowed the determination of high-resolution structures of the entire HCN1 (Burtscher et al., 2024; Lee & MacKinnon, 2017, 2019), and HCN4 proteins (Fig. 6). The latter has been solved with the pore in both the closed and in the open conformation (Saponaro, Bauer et al., 2021) and, more recently, with ivabradine inside the open pore (Saponaro et al., 2024). These structures have led to a more complete understanding of the protein architecture and gating movements, generating some obvious questions. What we want to address here is how does the HCN4 structure reproduce the original properties of the funny current? Does the new molecular insight validate or disprove the way features described decades ago have been interpreted? How do structural events recapitulate functional behaviour? As described below, the molecular properties of HCN4 channels faithfully confirm the original findings.

**What is the molecular pathway connecting S4 movement to pore opening?** Hyperpolarization-induced opening is one of the atypical properties of the $I_f$ current. When HCN channels were cloned, their expression in cell cultures immediately confirmed the basic $I_f$ properties, including activation by hyperpolarization, mixed cation permeability and cAMP potentiation of channel gating. Particularly puzzling was the realization that, similar to all Kv channels of the same superfamily, HCN channels also had a positively charged S4 voltage sensor (seven arginines and two lysines, a few more than in Kv channels (Chen et al., 2000). So, why should Kv channels open on depolarization, and HCN on hyperpolarization?

Indeed, it was shown that in HCN channels, membrane hyperpolarization actually moves S4 inwardly (Kasimova et al., 2019; Männikkö et al., 2002; Ramentol et al., 2020; Vemana et al., 2004) as in Kv channels (Vargas et al., 2012).

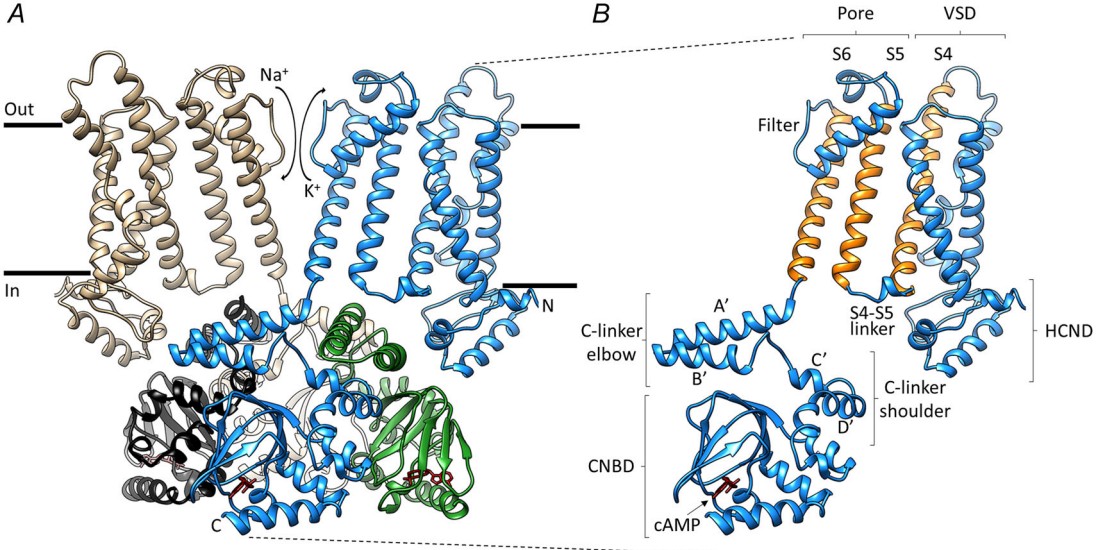

**Figure 6. Structural features of HCN4 channel**
*A*, cryo-EM Structure of the HCN4 channel bound to cAMP, in a cross-membrane view. For clarity, only two chains are shown in full (tan, blue), whereas, for the other two chains (black, green) only the cytosolic C-linker/CNBD domains are shown. *B*, protomer of HCN4 with the main structural elements indicated (reproduced with permission from Saponaro, Bauer et al., 2021).

This indicates that the inverted voltage dependency of HCN channels cannot be a reversal of the direction of the voltage-dependent movement of S4. The cause must therefore lie in a more complex rearrangement of S4 and/or in the coupling between voltage sensor movement and gating.

Early functional characterization of S4 movements in HCN channels led to the discovery that, during gating, the voltage sensor of HCN1 undergoes a rearrangement more complex than a simple inward movement (Bell et al., 2004). It was proposed that hyperpolarization causes the C-terminal cytosolic part of S4 to be exposed to a water-filled crevice, which collapses upon channel closure on depolarization. In the absence of available HCN structures, this complex motion of S4 could be only hypothesized as the result of a double rearrangement: a downward movement accompanied by either a rotation, or a swinging-tail motion of the S4 C-terminus.

A conclusive understanding of the S4 movement came thanks to the cryo-EM resolution-revolution of membrane protein structures. Indeed, the atomic model of HCN1 (Lee & MacKinnon, 2019) and HCN4 (Saponaro, Bauer et al., 2021) finally revealed the atomistic details of the voltage sensor architecture and the movements underlying channel opening. In particular, we refer to an HCN1 structure obtained with S4 forced into

the hyperpolarization-activated like position through chemical linkage (Lee & MacKinnon, 2019), as well as to an HCN4 structure where the loss of presumed lipids contacting S4 and S5 has induced a downward sliding of the voltage sensor and, consequently, pore opening (Saponaro, Bauer et al., 2021).

When compared with depolarization-activated Kv channels, HCN1 and HCN4 structures reveal two unpredicted peculiarities: a long S4 helix and a tight, parallel array of S4 and S5–S6 (pore) helices (Fig. 6B, orange helices). As for the voltage-dependent rearrangement and consequent pore opening, a detailed description of these events, as summarized in Fig. 7, can be derived by merging structural findings from recent cryo-EM data on HCN1 (Burtscher et al., 2024; Lee & MacKinnon, 2019) and HCN4 channels (Saponaro, Bauer et al., 2021; Saponaro et al., 2024).

As shown in Fig. 7A, S4 helix moves inwardly and, at the same time, breaks in two halves so that its C-terminal cytosolic part runs diagonally to the membrane surface, thus becoming partially solvent-exposed. This is in good agreement with the functional indication of a downward movement of S4 coupled with a water-filled crevice formation (Bell et al., 2004). By disrupting the parallel arrangement of S4, S5 and S6 (Fig. 7A, left), this peculiar S4 movement causes the loss of a highly relevant polar

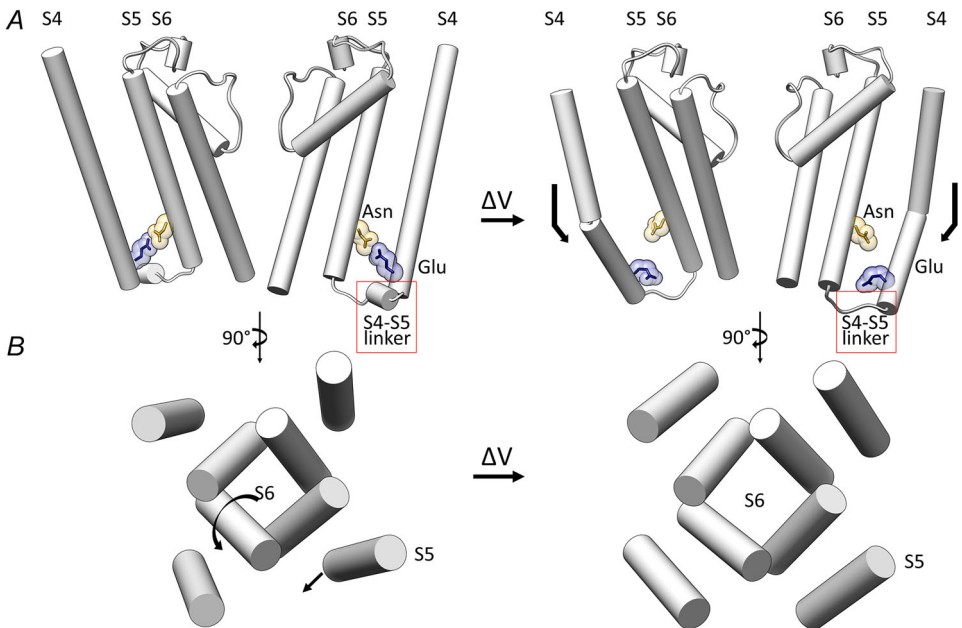

**Figure 7. Molecular steps of HCN channel opening**
*A*, opening is associated with inward sliding of S4 and concomitant breaking of its cytosolic C-terminal half caused by membrane hyperpolarization. These movements disrupt the planar array of S4, S5 and S6 helices, causing the unzipping of S4 and S5, the breaking of the hydrogen bond between glutamate (S5) and asparagine (S6) residues and the unfolding of the S4–S5 linker. *B*, the hyperpolarization-dependent movements of S4 causes upward/lateral tilting of S5, which then points out of the planar S4–S6 array. This S5 movement finally drives the rotation of the lower half of S6, leading to pore opening (from Lee & Mackinnon 2019; Saponaro, Bauer et al., 2021; Saponaro et al., 2024).

contact between a glutamic acid on S4 and an asparagine on S5 (Fig. 7*A*, right). Indeed, this hydrophilic interaction between S4 and S5 was highlighted by electrophysiology as part of the mechanism of channel gating (Chen, Mitcheson et al., 2001; Decher et al., 2004; Flynn & Zagotta, 2018; Ramentol et al., 2020).

Unexpectedly, the HCN1 pore solved by Lee & MacKinnon (2019) remained closed although S4 was in its hyperpolarization-activated state. This discrepancy was resolved in the HCN4 structure obtained in the presence of the membrane mimetic amphipols, which have chemically stabilized S5 splayed out (Saponaro, Bauer et al., 2021). In that structure, the loss of the tight interactions between S4 and S5 helices causes pore opening (Fig. 7*B*; see also Movie S1). Indeed, S5 movement drives a counterclock rotation of S6, leading to the widening of the intracellular pore cavity (Fig. 7*B*; see also Movie S1). Several molecular dynamics (MD) simulations performed with this HCN4 pore (Bauer et al., 2022; Krumbach et al., 2023; Saponaro, Bauer et al., 2021) confirmed that this indeed represents a conductive state displaying all functional features of HCN4 channels originally described for the native f-channels (see below). Furthermore, the recent resolution of the complex HCN4-ivabradine (Saponaro et al., 2024) reinforced the notion that the open-pore HCN4 structure is a reliable and powerful model for understanding permeation and block of these channels.

The comparison of open *vs.* closed HCN4 structures further revealed that pore opening is associated not only with the uplifting of S5, but also the unfolding of the S4–S5 linker (Fig. 7*A*, red square). The more recent cryo-EM structure of the open pore of both HCN1 (Burtscher et al., 2024) and HCN4 (Saponaro et al., 2024) confirmed that pore opening involves both occurrences. Note that all the structures described above were obtained at zero voltage, and thus voltage sensor movements and pore opening were chemically induced (Burtscher et al., 2024; Lee & MacKinnon, 2019; Saponaro, Bauer et al., 2021; Saponaro et al., 2024).

A similar gating model in which the upward tilt of the S5 N-terminus allows S6 the space needed for pore opening was also suggested by MD simulations of the HCN1 channel under hyperpolarization (Kasimova et al., 2019) and by transition metal fluorescence resonance energy transfer in sea urchin sperm HCN channels (Dai et al., 2019).

A third intriguing structural feature revealed by HCN4 structures is that the parallel arrangement of S4–S5–S6 helices is stabilized by lipids (Saponaro, Bauer et al., 2021), which allows to propose an active involvement of the membrane environment in HCN gating. Evidence for the involvement of lipids in channel gating agrees with functional studies investigating the role of membrane microdomains in the regulation of the voltage-dependency of HCN4 (Barbuti et al., 2004, 2007; Handlin & Dai, 2023).

**Structural basis for the mixed permeability to monovalent cations.** Given the high similarity of the primary amino acid sequence of HCN selectivity filter (SF) to that of selective $K^+$ channels, at a first glance, the mixed $Na^+/K^+$ permeability of HCN channels is inexplicable and thus particularly puzzling. The atomic model of HCN1 and particularly, the open pore of HCN4 have solved this apparent contradiction by showing that the difference in the permeation properties between HCN and $K^+$ channels does not depend on the primary sequence of the SF, but rather on the architecture and, most importantly, on the way dynamics is affected by the surrounding residues. Indeed, cryo-EM structures of HCN1 (Lee & MacKinnon, 2017) and of HCN4 channels (Saponaro, Bauer et al., 2021) have shown the presence of only two co-ordinating permeation sites in the pore, rather than the canonical 4 of Kv channels (Fig. 8*A*). More specifically, an elegant way to confirm a $Na^+$-$K^+$ mixed permeability has been the use of MD. *In silico* MD simulations (even if, as a result of computer-time limitations, still under conditions of large, non-physiological electrochemical gradients and on a very short time scale) actually reveal that both $K^+$ and $Na^+$ permeate through the filter not only because of the absence of all four canonical $K^+$ binding sites, but also because of the high flexibility of HCN SF compared to the more rigid filter of Kv channels (Fig. 8*B*) (Bauer et al., 2022; Krumbach et al., 2023; Saponaro, Bauer et al., 2021).

MD simulations in mixed $Na^+$-$K^+$ solutions showed weak $K^+$:$Na^+$ selectivity, with a preference of 4–6:1, in agreement with previous data from both native $I_f$ and heterologously expressed HCN4 currents (Ho et al., 1994; Lyashchenko & Tibbs, 2008; Moroni et al., 2000; Wollmuth & Hille, 1992), and thus provide an atomistic explanation of mixed permeability. MD showed a different permeation pathway for $K^+$ and for $Na^+$ and demonstrated that the 'knock-on' effect required for $Na^+$ permeation is provided by $K^+$. Early data had demonstrated that no inward funny current flows in the absence of extracellular $K^+$ ions. Following that original observation (DiFrancesco, 1982), more evidence had been collected showing that there is no or only limited conductance in a pure $Na^+$ solution (Edman & Grampp, 1989; Frace et al., 1992; Lyashchenko & Tibbs, 2008; Moroni et al., 2000; Solomon & Nerbonne, 1993). This is now fully reproduced with MD calculations. Indeed, there can never be more than a single $Na^+$ in the SF, as opposed to $K^+$ ions that alternate between a two-ion and single-ion configuration (Bauer et al., 2022; Krumbach et al., 2023; Saponaro, Bauer et al., 2021).

Another interesting observation on ionic permeability of HCN4 concerns the occurrence of a certain degree of adaptation to the permeating ion. Several

functional studies have described peculiar cation-specific permeation properties of HCN channels, such as a neglectable conductance to $Li^+$, an appreciable conductance to $Rb^+$, a weak sensitivity to $Ba^{2+}$ but a high sensitivity to $Cs^+$ (D'Avanzo et al., 2009; DiFrancesco, 1982; Ho et al., 1994; Moroni et al., 2000). The structure of the HCN4 open pore, synergistically combined with MD simulations, has now provided a reliable atomistic

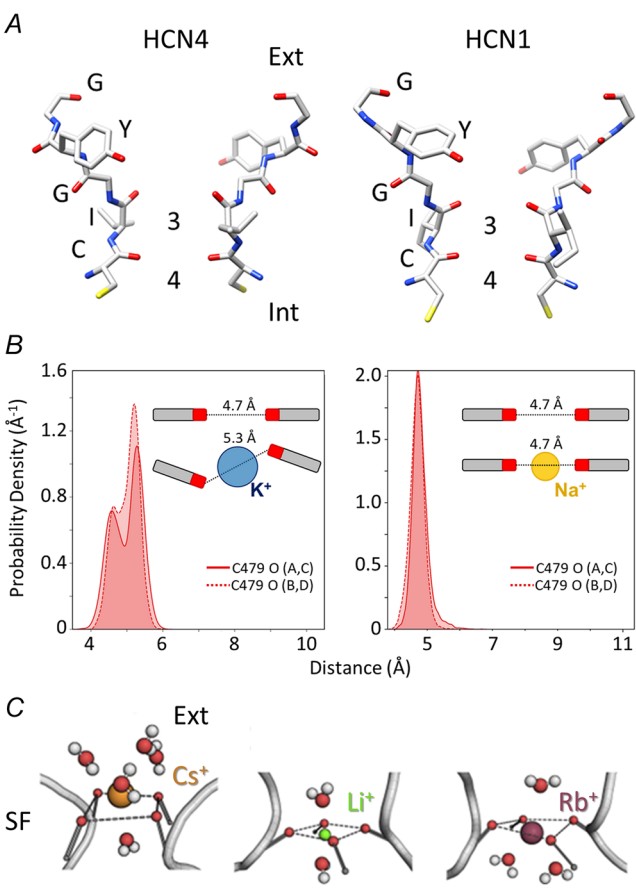

**Figure 8. Molecular basis of mixed $Na^+$–$K^+$ permeability**
*A*, selectivity filters of cryo EM-solved structures of HCN4 (Saponaro, Bauer et al., 2021) and HCN1 (Lee & MacKinnon, 2017). The sequence $^{479}$CIGYG$^{481}$ (rabbit HCN4 numbering) is marked (only two opposite subunits shown). In both cases, only two co-ordinating permeation sites are present (three and four following the canonical Kv channels numbering). *B*, density profiles of distance between opposite carbonyl oxygens of C479 in HCN4. Insets highlight C479 carbonyls distorted alignment in the presence of $K^+$ ions (from Saponaro, Bauer et al., 2021). *C*, representative snapshot of MD simulations showing $Cs^+$(orange), $Li^+$ (green) and $Rb^+$ (purple) within the SF of the HCN4 open pore, at the level of the carbonyl groups of I480 (only two opposite subunits of the SF are shown as grey ribbons). Carbonyl oxygens of the SF main chain co-ordinating the cations are shown as red spheres. Dashed grey lines indicate the oxygen plane of the co-ordinating carbonyl groups. Water molecules contributing to cation co-ordination are shown as red (oxygen) and white (hydrogen) spheres (reproduced with permission from Krumbach et al., 2023).

explanation of several different and peculiar permeation properties. Indeed, a high degree of flexibility of carbonyl oxygens of the filter forming the cation co-ordination sites (Fig. 8*B* and *C*) in synergy with $H_2O$ molecules, which support co-ordination (Fig. 8*C*), determines the specific biding/conduction properties of a given permeating cation.

A good example of how structural biology and *in silico* analysis can successfully interpret HCN biophysical properties is the atomistic explanation of a well-established feature of $I_f$ current, namely the voltage-dependent block exerted by external $Cs^+$ (DiFrancesco, 1982; Moroni et al., 2000). MD simulations of the HCN4 open pore in pure $Cs^+$ and in mixed $Cs^+/K^+$ show a peculiar transition pattern of $Cs^+$ that remains for a very long time in the SF, about six times that of $K^+$ (Krumbach et al., 2023). Because of this long residence in the SF, $Cs^+$ significantly decreases the frequency of $K^+$ transitions. This behaviour can account for the $Cs^+$ block of $I_f$ current. Interestingly, MD simulations also provide atomistic insights in other two intriguing properties of the $Cs^+$ action (DiFrancesco, 1982). First, although showing a strongly voltage-dependent block at negative voltages, at positive voltages, external $Cs^+$ activates $I_f$. Because of the $Cs^+$-dependent modification of filter geometry, $Cs^+$ occupation of SF induces conformational changes in the latter that favour $K^+$ release from to the extracellular side (Krumbach et al., 2023), a phenomenon that can explain the previously reported $Cs^+$-dependent increase of $I_f$ current at positive voltages (DiFrancesco, 1982). Second, application of a simple blocking model predicted that the electrical distance between $Cs^+$ bound in its blocking site and the outer membrane surface relative to the membrane thickness is independent of the $K^+$ concentration and has a value of ~0.7 (DiFrancesco, 1982; Moroni et al., 2000). Both these electrophysiological findings are now supported by MD simulations. Indeed, $Cs^+$ binding within the filter is independent from the presence or absence of $K^+$ and the estimation of value for its distance from the outer membrane surface is ~0.6 (Krumbach et al., 2023).

**Can we estimate the HCN4 single-channel conductance from the available structural information?** Single-HCN channel measurements have yielded conductance values close to that measured for f-channels in native SAN cells (~1 pS) (Fig. 3). For HCN2 channels, for example, values of 1.5, 1.67 and 2.5 pS have been reported from patch clamp recordings in HEK293 cells and oocytes, and from non-stationary analysis, respectively (Dekker & Yellen, 2006; Johnson & Zagotta, 2005; Thon et al., 2013).

A few studies of $I_f$ have reported conductance values more than one order of magnitude larger (Cámara-Checa et al., 2023; Michels et al., 2005). However, key features

typical of f/HCN channel kinetics are missing completely from the recordings shown, revealing that these records simply do not reflect the activity of f/HCN channels (Benndorf & DiFrancesco, 2024; DiFrancesco, 2005).

MD calculations allow evaluation of the single-channel conductance predicted for HCN4 channels in the open state. According to Saponaro, Bauer et al. (2021), nine $K^+$ ion passages are obtained with a simulation time of 2.1 µs. Given the elementary electric charge carried by each $K^+$ ion, the current transported is calculated to be 0.686 pA, which, in the presence of the applied voltage drop of 700 mV, leads to a single-channel conductance of 0.98 pS, which happens to be exactly the same value measured in the first recordings of the single f-channels (DiFrancesco, 1986) (Fig. 3).

**Do structural data explain the agonistic effect of cAMP?**
Following its original description (DiFrancesco & Tortora, 1991), the mechanism underlying cAMP-dependent activation first of the funny current, then of HCN channels, has turned out to be one of the most intriguing HCN properties that captured the attention of ion channel experts.

Several dedicated studies provided new insight into the processes of cAMP-dependent channel activation, both from functional and structural points of view, even before the HCN cryo-EM structures become available (Akimoto et al., 2014; Craven et al., 2008; Gross et al., 2018; Johnson & Zagotta, 2005; Lolicato et al., 2011; Pian et al., 2007; Porro et al., 2020; Saponaro et al., 2014, 2018; Wainger et al., 2001; Wang et al., 2001; Weissgraeber et al., 2017; Xu et al., 2010; Zagotta et al., 2003; Zhou & Siegelbaum, 2007).

As mentioned above, the key question addressed by these studies was how does cAMP binding partially remove the intrinsic C-terminus-associated self-inhibition of the channel? In the absence of available structures of full-length HCN proteins, these studies were limited to the cytosolic C-terminal portion of the channels and thus could not clarify how the latter interacts with the transmembrane region. Nonetheless, early work (Altomare et al., 2001; DiFrancesco, 1999) demonstrated that the cAMP-dependent depolarizing shift of the HCN channel open probability could be explained in terms of a facilitated binding of cAMP to channels in the open configuration (hyperpolarization). For example, an early study in native pacemaker cells supported evidence for an almost six-fold increase in the cAMP binding affinity for open *vs.* closed funny channels (DiFrancesco, 1999).

The study by DiFrancesco (1999) was in fact the first to propose a model based on an allosteric scheme describing the dual activation of funny channels by voltage hyperpolarization and cAMP. The model scheme assumed channels with two different configurations (in the paper termed 'subunits,' relaxed and tense, but in fact representing two channel 'states,' open and closed) to which cAMP could bind with different affinities. The model also assumed second-order Hodgkin–Huxley kinetics, equivalent to considering two structural channel subunits (instead of the existing four). Despite this approximation, the model was able to perfectly explain the basic aspects of voltage and cAMP dual modulation. The single, simple assumption of a cAMP binding affinity higher for open than for closed channels was sufficient in the model to generate the whole basic set of experimental findings: (1) a positive shift of the $I_f$ activaton curve; (2) a sigmoidal dependence of cAMP-induced shift against cAMP concentration; and (3) voltage shifts of activation and deactivation time constant curves. All experimental data could be correctly fitted. Fitting yielded a six-fold higher dissociation constants of cAMP binding to open (0.0732 µm) than to closed channels (0.4192 µm) and a shift of 13.7 mV for saturating concentrations of cAMP (DiFrancesco, 1999).

Years later, the augmented affinity of cAMP binding at negative voltages has been confirmed in a patch clamp fluorometry study where the hyperpolarization-dependent activation of HCN2 caused a three-fold enhancement of the channel affinity for a fluorescent cAMP analogue (Kusch et al., 2010). These data imply a coupling between two distant domains: the cytosolic, C-terminal CNBD and the transmembrane voltage sensor domain (VSD). Thus, it is to be expected that the molecular mechanism of HCN channel gating requires, at a given location, a short-range physical inter-action allowing the cross-talk of the two sensor domains CNBD and VSD. In this perspective, early functional studies with HCN1/2 chimeras indicated the existence of a physical association between the core transmembrane region and the cytosolic C-linker domain (Chen et al., 2007; Wang et al., 2001). Later studies have strengthened this concept by providing experimental and theoretical evidence for a mutual allosteric co-operation between CNBD and VSD with a mechanism that bypasses the pore (Hummert et al., 2018; Kusch et al., 2010; Wu et al., 2011).

The cryo-EM structures of HCN4 channels, supported by biochemical and biophysical results, have finally provided an exhaustive molecular understanding of the cAMP pathway, starting from the CNBD and propagating to the transmembrane region via a ligand-dependent association between the C-linker and the VSD (Saponaro, Bauer et al., 2021).

Figure 9*A* summarizes the cAMP-pathway as described by comparing the cAMP-free and bound structures of HCN4 protein: (1) cAMP binding to CNBD involves a complex system of rearrangements, corresponding to that already described for the isolated HCN2 CNBD (Saponaro et al., 2014), ultimately leading to an upward movement (black arrow) transmitted to the C-linker

domain immediately above; (2) the latter rearranges and propagates the cAMP signal via an extended surface of contacts between adjacent C-linker chains, also known as the mechanism of 'the elbow (A′-B′ helices) on shoulder (C′-D′ helices)' (Gross et al., 2018; Weissgraeber et al., 2017; Zagotta et al., 2003); (3) C-linker movements converge on A′-B′ helices that are reoriented relative to the above S4–S5 linker, the structural element bridging the cytosolic and the transmembrane region of the channel; and (4) A′-B′ helices and S4–S5 linker form a metal ion co-ordination site, termed the 'tetrad' (black dotted square) that ultimately allows the physical connection between CNBD and VSD.

As mentioned above, the structural study of HCN4 confirmed the previously described cAMP-dependent rearrangement of the cytosolic region, which can be then considered not only a general property of this channel family, but also a previously undescribed feature, namely the presence of a metal ion, co-ordinated by the tetrad, mediating the physical link between the C-linker and the S4–S5 linker. It is worth noting that the tetrad is not the exclusive pathway mediating the connection between cytosolic and transmembrane regions because its removal

does not completely abolish the cAMP effect on HCN4 (Saponaro, Sharifzadeh et al., 2021).

Full knowledge of the mechanical link between cytosolic and transmembrane regions of HCN channels will require further structural studies that are able to capture intermediate conformational states. A potential contributing mechanism has been proposed by functional investigation using rational mutagenesis on the three most studied HCN isotypes, 1, 2 and 4 (Porro et al., 2019). This study has revealed the existence of a common pathway of interaction between the CNBD and the VSD involving a newly resolved cytosolic 'HCN domain' (HCND) (Fig. 9*B*) at the channel N terminus (Lee & MacKinnon, 2017), a domain previously proposed to be important for channel assembly and trafficking (Tran et al., 2002).

Whether these or other mechanisms are common to all isotypes will clearly require fully detailed knowledge of the structures of HCN2 and HCN3 isotypes; recent cryo-EM resolution of HCN3 (Yu et al., 2024) indicates that, structurally speaking, HCN3 appears to be similar to HCN1.

The combination of structural and functional results can greatly contribute to solve the question of the

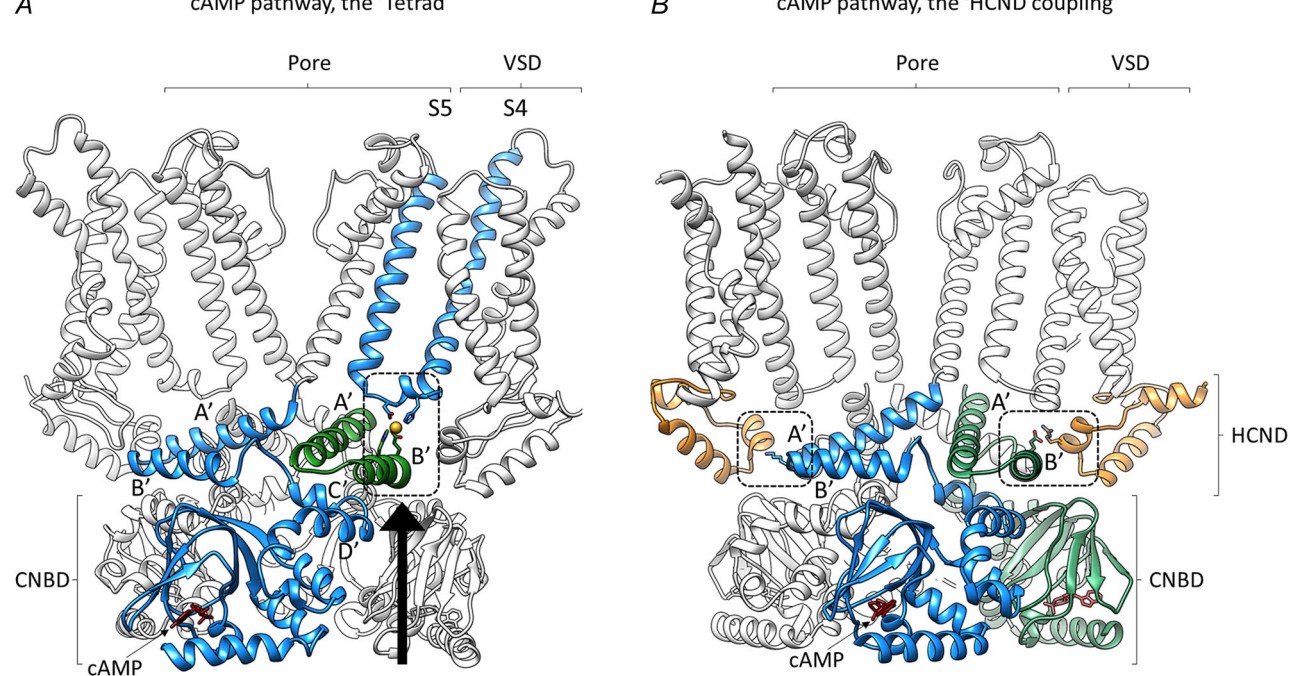

**Figure 9. Atomistic description of the cAMP pathway in HCN channels**
*A*, cAMP binding to HCN4 is associated with large conformational changes in the CNBD that are ultimately transmitted to the A′ and B′ helices ('the elbow') of the C-linker (black arrow) via contacts within (blue) and between (blue and green) C-linker chains. This readjusts the position of the C-linker elbow relative to the S4–S5 linker allowing tetrad formation (circled with a broken line, metal ion in yellow). The tetrad physically links the cytosolic regulatory region with the transmembrane one (from Saponaro, Bauer et al., 2021). *B*, HCN1 structure bound to cAMP; in this structure, a network of contacts (circled with broken lines) was identified between C-linker elbows (blue and green) and HCNDs (orange) that propagates cAMP signalling from the cytosolic region to the transmembrane one in HCN1, 2 and 4 channels (from Porro et al., 2019).

long-range allosteric connection between the cytosolic regulatory tail and the voltage sensing machinery, thus providing a molecular interpretation of the mutual allosteric co-operation between voltage and cAMP initially assumed to explain experimental electrophysiological data (Altomare et al., 2001; DiFrancesco, 1999; Wainger et al., 2001; Wang et al., 2001). Nonetheless, it is worth noting that a limitation for all the mechanisms described above is that they are based on structures solved at 0 mV, a condition where the voltage sensor cannot exert its hyperpolarization-dependent facilitatory effect on the cAMP binding affinity (Altomare et al., 2001; DiFrancesco, 1999; Hummert et al., 2018). Moreover, the only structural attempts to capture voltage sensor in the activated conformation come from studies of HCN1 (Burtscher et al., 2024; Lee & MacKinnon, 2019), an isotype in which the extent of cAMP potentiation is smaller than in HCN2 and HCN4 (Altomare et al., 2003; Wainger et al., 2001). Because of this, the cAMP-induced rearrangements in these structures were limited to the CNBD, as expected. From the above, further structural and functional efforts are therefore needed on the more cAMP-sensitive isotypes HCN2 and HCN4 to properly locate the above-described allosteric pathways of communication between the CNBD and the VSD in the context of the channel activation by hyperpolarization.

Still considering cAMP regulation, the cryo-EM structures of HCN channels have also contributed to the discovery that this family of channels is endowed with an affinity switch for cAMP (Porro et al., 2024). The newly described helices D and E ($\alpha$DE), downstream of the CNBD, form a helix-turn-helix motif that binds to and stabilizes the CNBD in a high affinity state for cAMP. The presence of the helices increases cAMP efficacy by 30-fold in patch clamp experiments and in measurements from isolated CNBD fragments by isothermal titration calorimetry. The results further highlight that $\alpha$DE helices interact with the same element of the CNBD that is the target of another controller of HCN cAMP affinity, the brain regulatory protein TRIP8b (Santoro et al., 2004). These cryo-EM-based results highlight a modulating mechanism based on changes in binding affinity, rather than changes in cAMP concentration. The study by Porro et al. (2024) demonstrated that helices D and E are not part of the allosteric communication pathway between VSD and CNBD because the fold change in the affinity induced by the hyperpolarization (i.e. VSD movement) is preserved upon removal of the two helices. Therefore, helices D and E probably represent an intrinsic intramolecular mechanism controlling cAMP binding affinity.

**Can structural data elucidate the mechanism of ivabradine block?** Because ivabradine is an HCN4 open pore blocker, the channel needs to be opened to access its binding site in the pore cavity. The availability of an HCN4 pore in the open conformation (Saponaro, Bauer et al., 2021; Saponaro, Sharifzadeh et al., 2021) has made it possible to solve the cryo-EM structure of the channel with ivabradine bound inside its intracellular pore vestibule (Saponaro et al., 2024).

This latter study unambiguously confirmed that the drug is an open-channel blocker and validated, at an atomistic level, a large set of functional data describing ivabradine block as use and state-dependent (Bois et al., 1996; Bucchi et al., 2002, 2006; DiFrancesco, 1994; Thollon et al., 1994).

The bulky aromatic components of the drug develop extensive apolar contacts with a tyrosine and an isoleucine in the HCN4 pore vestibule, both residues previously proposed as part of the drug binding site (Bucchi et al., 2013), but also with a cysteine in the SF, a residue that contributes to the dynamics of ivabradine block via apolar interactions. Note that Cys is peculiar of HCN channels because all other $K^+$ channels, including CNG and NaK channels, have a highly conserved Ser/Thr in the equivalent position (Liu & Lockless, 2013). Moreover, the structure clarifies the role of a fourth residue, a phenylalanine, previously proposed as part of the drug contacts inside the pore (Bucchi et al., 2013). The cryo-EM of HCN4 instead revealed that this residue does not point inside the pore, and therefore does not directly contact the drug. Nonetheless, phenylalanine is important because it controls indirectly ivabradine inside the pore vestibule via an edge-to-face $\pi$-interaction with the above-mentioned tyrosine. This indirect influence exerted by phenylalanine on the drug, together with the fact that its side chain points toward the S6–S5 interface where a lipid moiety is found in the closed HCN4 pore structure (Saponaro, Bauer et al., 2021), raises the question of whether the presence of lipids surrounding the pore can affect ivabradine binding, perhaps via long-range allosteric interactions.

Given the large space occupied by the drug inside the pore vestibule and the presence of a charged tertiary amine located just below the SF, two possible modalities of block were considered: pore hindrance and electrostatic interaction. Once again, MD worked synergistically with structural data by providing the unique opportunity to discriminate between these hypotheses. Simulations performed with the protonated and neutral ivabradine allowed verification that the block is electrostatic and depends on the mutual repulsion between the charged amine of the drug and the cation in the SF (Fig. 10). These data fully validate, and provide an elegant molecular interpretation of, the 'current-dependent' block model previously proposed in studies of native $I_f$ and HCN4 current (Bucchi et al., 2002, 2006, 2013).

It is also worth noting that a cryo-EM structure of HCN1 with the closed pore and ivabradine bound outside

has been recently solved (Che et al., 2024). This structure validates a previously proposed HCN1-specific blocking model where this HCN subtype is blocked also in the closed configuration (closed-channel block) (Bucchi et al., 2006). This new structural information, together with the identical pore cavity sequence among HCN subtypes, predicts that subtype-specific drugs will probably not emerge from chemical modifications of ivabradine binding inside the pore. In alternative to studies of inside-binding molecules, it will be important to explore molecules that are able to dock outside the pore, where subtype specificity has been demonstrated by cryo-EM studies (Che et al., 2024; Saponaro, Bauer et al., 2021).

## Conclusions

The atomic resolution structures of HCN channels, obtained via cryo-EM, allow exploration of structural aspects with an exploded view and reveal the molecular details that underlie the macroscopic properties and functional behaviour of native f-channels in cardiac cells. Further structural studies will also help to unravel the basis for the differences among the various HCN orthologs expressed in non-mammalian organisms and in non-excitable cells, as well as understand how these differences translate into specific functional behaviour.

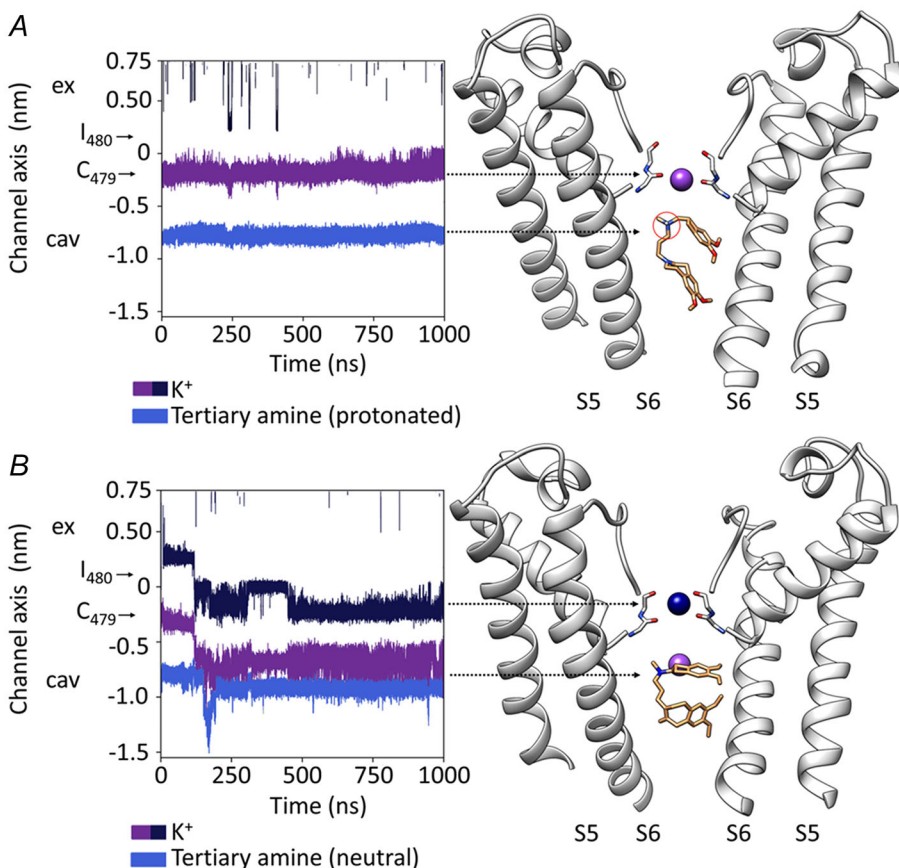

**Figure 10. Molecular mechanism of ivabradine block of the open HCN4 pore**
One microsecond MD simulations of ivabradine with its tertiary amine in the protonated (*A*) and neutral (*B*) form. Left: trajectories for $K^+$ ions (purple and dark blue) in the SF and for the tertiary amine of ivabradine (light blue) in the open pore cavity of HCN4. Carbonyl groups of C479 and I480, which form the permeation pathway, are indicated by black arrows. Ex, extracellular side of the pore; cav, intracellular pore cavity. Right: representative snapshots of the MD simulations shown on the left. Arrowheads link trajectories on the left with corresponding atoms on the right. Only two opposite chains of the pore are shown. *A*, in normal conditions, the charged ivabradine (the protonated tertiary amine is circled in red) is stably located in the intracellular cavity, just below the SF. This blocks $K^+$ (purple) in the SF. *B*, when the tertiary amine of ivabradine is neutralized, the drug still occupies the cavity but with a different orientation, not interfering with a $K^+$ binding site, and $K^+$ permeation occurs. Indeed, a second $K^+$ ion (dark blue) enters from the extracellular side of the SF and displaces the second ion (purple) into the intracellular pore cavity (reproduced with permission from Saponaro, Bauer et al., 2021).

Thus far, most of the original properties of funny channels have found a molecular explanation. Structural details of the HCN proteins elegantly explain and faithfully 'mirror' the functional properties of the funny current as described in the original experiments.

Achieving full confirmation of old measurements and validation of the original interpretation of experimental data by new molecular findings is, to old experimentalists (and young experimentalists alike), one of the most gratifying outcomes of research activity.

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

## Additional information

### Competing interests

The authors declare that they have no competing interests.

### Author contributions

Both authors contributed equally to the study conceptualization, as well as writing and revising the manuscript. All persons designated as authors qualify for authorship, and all those who qualify for authorship are listed.

### Funding

The work was supported by the Fondation Leducq Research Grant no. 19CVD03 to DD.

### Acknowledgements

We thank Anna Moroni, Gerhard Thiel and Alessandro Porro for discussions.

Open access publishing facilitated by Universita degli Studi di Milano, as part of the Wiley - CRUI-CARE agreement.

### Keywords

cryo-EM, funny current, HCN channels, pacemaker

### Supporting information

Additional supporting information can be found online in the Supporting Information section at the end of the HTML view of the article. Supporting information files available:

**Peer Review History**
**Movie S1**

