## [Peer Review History · The Journal of Physiology]

Structure mirroring function: What's the "matter" with the funny current?

Dario DiFrancesco and Andrea Saponaro
DOI: 10.1113/JP287209

Corresponding author(s): Dario DiFrancesco (dario.difrancesco@unimi.it)

The following individual(s) involved in review of this submission have agreed to reveal their identity: Ming Lei (Referee #1)

Review Timeline:

Submission Date:	06-Nov-2024
Editorial Decision:	19-Dec-2024
Revision Received:	20-Jan-2025
Accepted:	03-Feb-2025

Senior Editor: Laura Bennet

Reviewing Editor: T Alexander Quinn

Transaction Report:

Dear Dr DiFrancesco,

Re: JP-TR-2024-287209 "Structure mirroring function: What's the "matter" with the funny current?" by Dario DiFrancesco and Andrea Saponaro

Thank you for submitting your manuscript to The Journal of Physiology. It has been assessed by a Reviewing Editor and by 2 expert referees and we are pleased to tell you that it is potentially acceptable for publication following satisfactory major revision.

Please address all the points raised and incorporate all requested revisions or explain in your Response to Referees why a change has not been made. We hope you will find the comments helpful and that you will be able to return your revised manuscript within 9 months. If your article is for a Special Issue, please note that we require your revised version within 2 months (rather than 9 months) in order to keep the Special issue on track. If you require longer than this, please contact journal staff: jp@physoc.org. Please note that this letter does not constitute a guarantee for acceptance of your revised manuscript.

ABSTRACT FIGURES: Authors are expected to use The Journal's premium BioRender account to create/redraw their Abstract Figures. Information on how to access this account is here:

<https://physoc.onlinelibrary.wiley.com/journal/14697793/biorender-access>.

REVISION CHECKLIST:

IMPORTANT POINTS TO NOTE WHEN REVISING YOUR MANUSCRIPT:

We look forward to receiving your revised submission.
If you have any queries, please reply to this email and we will be pleased to advise.

Yours sincerely,

Laura Bennet
Senior Editor
The Journal of Physiology

REQUIRED ITEMS

- Please include an Abstract Figure file, as well as the Figure Legend text within the main article file. The Abstract Figure is a piece of artwork designed to give readers an immediate understanding of the Review Article and should summarise the main conclusions. If possible, the image should be easily 'readable' from left to right or top to bottom. It should show the physiological relevance of the Review so readers can assess the importance and content of the article. Abstract Figures should not merely recapitulate other figures in the Review. Please try to keep the diagram as simple as possible and without superfluous information that may distract from the main conclusion of the Review. Abstract Figures must be provided by authors no later than the revised manuscript stage and should be uploaded as a separate file during online submission labelled as File Type 'Abstract Figure'. Please ensure that you include the figure legend in the main article file. All Abstract Figures will be sent to a professional illustrator for redrawing and you may be asked to approve the redrawn figure before your paper is accepted.

- Your MS must include a complete "Additional information section" with the following 4 headings and content:

Competing Interests: A statement regarding competing interests. If there are no competing interests, a statement to this effect must be included. All authors should disclose any conflict of interest in accordance with journal policy.

Author contributions: Each author should take responsibility for a particular section of the study and have contributed to writing the paper. Acquisition of funding, administrative support or the collection of data alone does not justify authorship; these contributions to the study should be listed in the Acknowledgements. Additional information such as 'X and Y have contributed equally to this work' may be added as a footnote on the title page.

It must be stated that all authors approved the final version of the manuscript and that all persons designated as authors qualify for authorship, and all those who qualify for authorship are listed.

Funding: Authors must indicate all sources of funding, including grant numbers. If authors have not received funding, this must be stated.

It is the responsibility of authors funded by RCUK to adhere to their policy regarding funding sources and underlying research material. The policy requires funding information to be included within the acknowledgement section of a paper. Guidance on how to acknowledge funding information is provided by the Research Information Network. The policy also requires all research papers, if applicable, to include a statement on how any underlying research materials, such as data, samples or models, can be accessed. However, the policy does not require that the data must be made open. If there are considered to be good or compelling reasons to protect access to the data, for example commercial confidentiality or legitimate sensitivities around data derived from potentially identifiable human participants, these should be included in the statement.

Acknowledgements: Acknowledgements should be the minimum consistent with courtesy. The wording of acknowledgements of scientific assistance or advice must have been seen and approved by the persons concerned. This section should not include details of funding.

- The reference list must be in alphabetical order, rather than numbered, to comply with our Journal format.

- Please upload separate high quality figure files via the submission form.

- Please provide high quality and high resolution images and figures.

- Author profile(s) must be uploaded via the submission form. Authors should submit a short biography (no more than 100 words for one author or 150 words in total for two authors) and a portrait photograph of the two leading authors on the paper. These should be uploaded and clearly labelled together in a Word document with the revised version of the manuscript. Any standard image format for the photograph is acceptable, but the resolution should be at least 300 DPI and preferably more. A group photograph of all authors is also acceptable, providing the biography for the whole group does not exceed 150 words.

- It is the authors' responsibility to obtain any necessary permissions to reproduce previously published material and to list these within the main article file. For information, please see: https://jp.msubmit.net/cgi-bin/main.plex?form_type=display_requirements#permissions.

- Please include a full title page as part of your main article (Word) file, which should contain the following: title, authors, affiliations, corresponding author name and contact details, keywords, and running title.

- Please ensure that the Article File you upload is a Word file.

EDITOR COMMENTS

Reviewing Editor:

Your paper has been reviewed by two experts in the field. They felt it is a well written review that nicely covers the subject and will ultimately be impactful in the field. While the first reviewer had only one minor comment to address, the second reviewer has made multiple useful suggestions that will help improve the overall clarity, accuracy, and comprehensiveness of the manuscript. Please submit a revision that addresses these comments, along with a point-by-point response to the reviewers.

Please also see 'Required Items' above.

REFEREE COMMENTS

Referee #1:

This is a very well written review focusing on the structure-function relationship. Congratulations to the authors.

I have only one comment for the authors to consider:

In the section page Selective blockers of funny channels as heart rate-reducing agents (page 9), can authors also discuss if it can be a new target for antiarrhythmic treatment as proposed by Modernized Classification of Cardiac Antiarrhythmic Drugs (Lei et al Circulation, 2018).

Referee #2:

Saponaro and Difrancesco

Summary

This review is focused on the funny current of the heart and recent reports of structures for the subunits that form it. When initially studied, many of the features of the currents were unusual and hard to understand. Here, the authors present an interesting and necessary synthesis of the properties of the funny current and how they can be understood in the context of the more recent structures of HCN isoforms. I think that the inclusion of early studies of the funny current are critical and a strength of this review. These studies are not always cited or referred to appropriately which is unfortunate for at least two reasons. First, as mentioned here, many of the characteristics of the funny current are well-explained by the function and structures of the HCN isoforms. Second, I also think that some features detailed in early studies do not necessarily fit with what is known currently about the isoforms. This leads me to ask if it might be useful for the review to address some of the original findings that are not necessarily explained by structural and functional information derived from the four mammalian HCN isoforms? This might be helpful for understanding what types of questions and experiments could be considered for future studies.

I have some comments on the form of the review and some of the information which was not always clear to me. Some of the writing and grammar were likewise not always clear.

Introduction

This section nicely introduces the topic of the review.

In the middle of page 2,

"HCN channels belong to the superfamily of the pore-loop cation channels."

This is a large family and not everyone who reads this might have a good sense of what they are? Could you briefly talk about this group and possibly provide a reference?

From pore-loop cation channels the introduction moves to the relationship between HCN and Kv and CNG channels.

On page 2, "Like Kv and CNG channels, HCN channels are tetramers, but they have some distinctive properties. For example, Kv channels are gated by voltage, and CNG channels by cyclic nucleotides, while HCN channels are dually gated by both. Also, curiously, HCN channels open on hyperpolarization, not on depolarization like Kv channels, despite structural similarities."

What is the relationship? Are all pore-loop channels tetramers? There are some Kv channels that do open upon hyperpolarization such as the KAT1 channel in plants. KAT1 channels also possess a cyclic nucleotide-binding domain and are evolutionarily close to HCN channels which suggests that the information derived from the HCN structures could apply to an even wider group of proteins. Overall, would it be helpful to expand this piece in order to place the HCN channel in context of other channel families, including Kv and CNG channels?

At the top of page 3,

"As we discuss below, structural details of the HCN channel proteins elegantly explain and faithfully "mirror" the functional properties of the funny current as described in original experiments."

I think that this is true for many features of the funny current and cloned channel isoforms. But I think there may also be features of the funny current that are not completely explained by the structures and vice versa? This may be where future studies could be aimed?

Early description of the If properties

This section summarizes the early work on the funny current. At the beginning, the text implies that the work to be discussed is from pacemaker cells isolated from the sinoatrial node. However, there are data on funny current properties which are obtained in Purkinje fibres? e.g. Figure 2. Some clarification or description of experiments in Purkinje fibres might also be helpful?

The second paragraph of page 4,

"The proposal of a novel If -based pacemaker model had an impact in cardiac physiology because it provided a new mechanism able to not only initiate pacemaker activity, but also to control pacemaker frequency and to mediate sympathetic rate modulation"

The main ideas in this sentence appear to be stated twice in a row in slightly different form?

At this point, it might be useful to mention that funny current is also important for parasympathetic rate modulation? eg "to mediate sympathetic and parasympathetic modulation"

I am not sure it has been shown that the funny current mediates sympathetic modulation of rate but "contributes to" might be a better term than mediates?

Another section appears here,

"HCN channels are the alpha subunits of native funny channels".

My guess is that this is not part of the section titled "Early description of If properties" but this is not clear as the sections are not organized by letters or by numbers? It would seem that this is a different section altogether as it outlines the cloning of HCN channel subunits.

Although there is a brief description of 4 isotypes (isoforms is usually used?), it is not clear from the description here that these come from four separate genes? There is only a brief mention that they are distributed in heart and brain, and that the HCN4 isoform predominates in the SA node. There is no mention of the other isoforms that are found in the heart and regions other than the SA node such as in Purkinje fibres, where much of the early work on If was carried out and which is discussed here in the section on early description of If properties. In Purkinje fibres, there is some evidence that HCN2 is the

main isoform, which behaves functionally in a similar but not identical way with HCN4 in overexpression systems.

There is a very brief piece in this section on functional properties where experiments on the effect of ivabradine on HCN4 is mentioned. Not sure why this information is found in this section?

It may also be important that even though HCN4 is the main isoform (gene) in the SA node, there are key differences in the function of the funny current of the SA node and currents produced by overexpression of the HCN4 channel. For example, I think the funny current produced in heterologous cells that express HCN4 open and close more slowly than the funny current of sinoatrial pacemaker cells? An interesting idea that HCN1 may be expressed in the node and contribute to those differences has been proposed by the DiFrancesco group.

This section seems like it is hanging as much of the functional experiments are discussed in the later section on structure. Is the purpose of this section to introduce the cloning of the four isoforms? Overall, I am not clear on the purpose of this section?

For cloning HCN, I noted that papers by Santoro et al and Ludwig et al are cited here. However, there is another paper on HCN that was published back-to-back with the Ludwig paper in the same issue of Nature by Gauss et al, which you could also cite.

Structure mirroring function

This is the main section that connects original studies with the recent work on structures.

Page 11. "Does the new molecular insight validate or disprove the features described over four decades ago?" I am not sure that the early experimental findings would be invalidated or disproved by more recent work? I think they inform each other?

"On page 11 As is described below, the molecular properties of HCN4 channels faithfully confirm original findings." As I mention above, I think that some of the interesting observations of the original studies are not readily explained by the cloning, functional expression and structure? So, I am not sure that they faithfully confirm original studies. I think that this also suggests that the structure confirms function whereas I think this works both ways?

In the second paragraph of page 11, "More recently, the cryo-EM technique has allowed to solve high-resolution structures of the entire HCN1 (Lee and MacKinnon, 2017, 2019; Burtscher et al., 2024). "

I am not sure the entire structure was used in either study? My understanding is that there were pieces removed from the HCN1 channel perhaps to improve expression? Is this also true for HCN4 and HCN3?

In the structure of HCN4 solved by cryo-EM, an open channel is also thought to result as well as a closed channel. Presumably, the cross-membrane voltage is 0 mV in the cryo-EM structure. I think this may require some explaining as there is also functional data suggesting the channels may remain open at depolarized voltages? It also raises the question of what happens to the pore when the voltage sensors move upon hyperpolarization? Does it just further facilitate opening or stabilize the open conformation?

On page 14,

"The atomic model of HCN1 and, in particular, the open pore of HCN4, have solved this apparent contradiction by showing that the difference in the permeation properties between HCN and K⁺ channels does not depend on the primary sequence, but rather on the architecture and, most importantly, on the dynamics of their respective SFs."

This is a bit confusing because, ultimately, I think the architecture and dynamics do depend on the primary sequence. So, I am not quite sure what this phrase is intending?

Page 19. "Thus, it is to be expected that the molecular mechanism of HCN channel gating requires, at a given location, a short-range physical interaction allowing the crosstalk of the two sensor domains CNBD and VSD."

I think this is an interesting finding but I am not sure it is expected because both the voltage sensing regions and CNBD are attached to the pore? What does a more direct connection achieve than an indirect one does not?

On page 20. It is mentioned that the HCN1 isoform is insensitive to cAMP, which may not be clear. I think the maximum effect of cAMP is smaller than for the HCN2 and HCN4 isoform. However, there is data that suggests the potency of cAMP is similar between HCN1 and HCN2 (I think in reference 94).

Page 10. Two papers are cited that reported the first cloning of HCN subunits. On page 11, sensitivity to cAMP is again used. Perhaps, it needs to be defined?

Page 17 "Both these electrophysiological findings are now supported by MDs". MD simulations?

Page 19. "All experimental data could be faultlessly fitted" perhaps well-fitted?

Page 20 "a newly defined cytosolic "HCN domain" (HCND, Fig. 9B), at the channel N-terminus (Lee and MacKinnon, 2017)."
This domain was actually noted a few years ago but its structure was solved in 2017.

END OF COMMENTS

EDITOR COMMENTS

Reviewing Editor:

Your paper has been reviewed by two experts in the field. They felt it is a well written review that nicely covers the subject and will ultimately be impactful in the field. While the first reviewer had only one minor comment to address, the second reviewer has made multiple useful suggestions that will help improve the overall clarity, accuracy, and comprehensiveness of the manuscript. Please submit a revision that addresses these comments, along with a point-by-point response to the reviewers.

Please also see 'Required Items' above.

Thank you. We have addressed all the points raised by the reviewers and amended the text where required. This has involved modifying, inserting, removing text as according to the reviewers' comments, which as a consequence has changed somewhat the length of the manuscript and the number of references. To improve clarity, we have numbered the multiple points raised by reviewer #2 and provided point-by-point answers; answers are in Italic font.

We have also inserted an Abstract figure as required. We are uploading two versions of the manuscript, one with changes highlighted and one clean with no changes tracked, and separate high-resolution figure files, and supporting information (Video).

We hope this is satisfactory.

REFEREE COMMENTS

Referee #1:

This is a very well written review focusing on the structure-function relationship. Congratulations to the authors.

I have only one comment for the authors to consider:

In the section page Selective blockers of funny channels as heart rate-reducing agents (page 9), can authors also discuss if I_f can be a new target for antiarrhythmic treatment as proposed by Modernized Classification of Cardiac Antiarrhythmic Drugs (Lei et al Circulation, 2018).

Thank you for your congratulations and for your comment.

We have now mentioned that the Modernized Classification of Cardiac Antiarrhythmic Drugs of Lei et al, 2018 has recently proposed that I_f inhibitors like ivabradine represent a new antiarrhythmic class (class 0), and has identified the I_f reduction as a new target for antiarrhythmic treatment.

Referee #2:

Saponaro and Difrancesco

1. Summary

This review is focused on the funny current of the heart and recent reports of structures for the subunits that form it. When initially studied, many of the features of the currents were unusual and hard to understand. Here, the authors present an interesting and necessary synthesis of the properties of the funny current and how they can be understood in the context of the more recent structures of HCN isoforms. I think that the inclusion of early studies of the funny current are critical and a strength of this review. These studies are not always cited or referred to appropriately which is unfortunate for at least two reasons. First, as mentioned here, many of the characteristics of the funny current are well-explained by the function and structures of the HCN isoforms. Second, I also think that some features detailed in early studies do not necessarily fit with what is known currently about the isoforms. This leads me to ask if it might be useful for the review to address some of the original findings that are not necessarily explained by structural and functional information derived from the four mammalian HCN isoforms? This might be helpful for understanding what types of questions and experiments could be considered for future studies.

I have some comments on the form of the review and some of the information which was not always clear to me. Some of the writing and grammar were likewise not always clear.

Thank you for your general comments. In this review we have taken advantage of recent knowledge of atomistic details of HCN channels structure to revisit, and attempt to find a molecular basis for, known funny current properties, as originally described in early experiments, specifically in cardiac cells.

We are aware that several relevant functional aspects of the funny current still remain to be addressed from a structural viewpoint. We are also aware that properties of channels belonging to the same superfamily that are not expressed in mammals, such as for example SpIH channels, differ from those of HCN channels in many aspects, such as the transient activation in the absence of cAMP, the large increase in the presence of cAMP, the lack of cGMP modulation. Dealing with these differences is beyond the scope of our review, since we have specifically focussed our attention on the cardiac funny current.

On the other hand, we share the reviewer's view that future structural studies will be useful to provide important information to unravel the basis for these differences, and to understand how these translate into functional behaviour. We have now inserted quotations of studies of non-mammalian isoforms and mentioned more clearly the above concept in the Conclusions.

Introduction

This section nicely introduces the topic of the review.

2. In the middle of page 2,

"HCN channels belong to the superfamily of the pore-loop cation channels."

This is a large family and not everyone who reads this might have a good sense of what they are? Could you briefly talk about this group and possibly provide a reference?

From pore-loop cation channels the introduction moves to the relationship between HCN and Kv and CNG channels.

On page 2, "Like Kv and CNG channels, HCN channels are tetramers, but they have some distinctive properties. For example, Kv channels are gated by voltage, and CNG channels by cyclic nucleotides, while HCN channels are dually gated by both. Also, curiously, HCN channels open on hyperpolarization, not on depolarization like Kv channels, despite structural similarities."

What is the relationship? Are all pore-loop channels tetramers? There are some Kv channels that do open upon hyperpolarization such as the KAT1 channel in plants. KAT1 channels also possess a cyclic nucleotide-binding domain and are evolutionarily close to HCN channels which suggests that the information derived from the HCN structures could apply to an even wider group of proteins. Overall, would it be helpful to expand this piece in order to place the HCN channel in context of other channel families, including Kv and CNG channels?

Thank you for these comments. In the first version we opted for a concise description also because of space limitations, but we agree that more detailed information on the superfamily of pore-loop channels is useful. We have now clarified the meaning of pore-loop channels and explained that HCN, Kv and CNG channels belong to this superfamily.

For the sake of completeness, we have also mentioned the only two Kv channels presently known to open on hyperpolarization: KAT1 from Arabidopsis thaliana (Latorre et al., 2003), and the archaeobacterial MVP (Sesti et al., 2003).

3. At the top of page 3,

"As we discuss below, structural details of the HCN channel proteins elegantly explain and faithfully "mirror" the functional properties of the funny current as described in original experiments."

I think that this is true for many features of the funny current and cloned channel isoforms. But I think there may also be features of the funny current that are not completely explained by the structures and vice versa? This may be where futures studies could be aimed?

Yes, see above the answer to point 1.

4. Early description of the I_f properties

This section summarizes the early work on the funny current. At the beginning, the text implies that the work to be discussed is from pacemaker cells isolated from the sinoatrial node. However, there are data on funny current properties which are obtained in Purkinje fibres? e.g. Figure 2. Some clarification or description of experiments in Purkinje fibres might also be helpful?

Thank you for this comment. Several original I_f properties were indeed first described in Purkinje fibres, and since investigation of I_f in Purkinje fibres had to wait the re-interpretation of the previous pacemaker "current" I_{K2} model, we thought it appropriate to refer to Purkinje fibres' data

only after mentioning the re-interpretation. However, the referee is correct in noticing that we missed to mention this in the first paragraph of the section “Early description of the I_f properties”. We have now reworded the sentence to state that the elementary properties of I_f were also described in Purkinje fibres after the I_{K2} re-interpretation.

5. The second paragraph of page 4,

"The proposal of a novel I_f -based pacemaker model had an impact in cardiac physiology because it provided a new mechanism able to not only initiate pacemaker activity, but also to control pacemaker frequency and to mediate sympathetic rate modulation"

The main ideas in this sentence appear to be stated twice in a row in slightly different form?

Thank you. The above sentence has been removed.

6. At this point, it might be useful to mention that funny current is also important for parasympathetic rate modulation? eg "to mediate sympathetic and parasympathetic modulation"

I am not sure it has been shown that the funny current mediates sympathetic modulation of rate but "contributes to" might be a better term than mediates?

“mediating” replaced with “contributing to mediate”

7. Another section appears here,

"HCN channels are the alpha subunits of native funny channels".

My guess is that this is not part of the section titled "Early description of I_f properties" but this is not clear as the sections are not organized by letters or by numbers? It would seem that this is a different section altogether as it outlines the cloning of HCN channel subunits.

We had originally chosen the section “Structure mirroring function” as the one where to address specific cryo-EM-related structural data. Anyway we agree that the sub-section “HCN channels are the alpha subunits of native funny channels” can also suitably fit into that section. We have therefore moved it ahead.

8. Although there is a brief description of 4 isotypes (isoforms is usually used?), it is not clear from the description here that these come from four separate genes? There is only a brief mention that they are distributed in heart and brain, and that the HCN4 isoform predominates in the SA node. There is no mention of the other isoforms that are found in the heart and regions other than the SA node such as in Purkinje fibres, where much of the early work on I_f was carried out and which is discussed here in the section on early description of I_f properties. In Purkinje fibres, there is some evidence that HCN2 is the main isoform, which behaves functionally in a similar but not identical way with HCN4 in overexpression systems.

*We have clarified that in humans, the four protein members of the HCN channel family (HCN1-4) are encoded by four distinct genes (*hcn1-4*). This is the reason why we used the term “isotypes” (homolog proteins encoded from distinct but homolog genes) and not “isoforms” (homolog proteins originated by slicing variants of the same gene).*

Moreover, we have better clarified their pattern of expression, with emphasis on the expression in sinoatrial node (SAN) cells.

9. There is a very brief piece in this section on functional properties where experiments on the effect of ivabradine on HCN4 is mentioned. Not sure why this information is found in this section?

We have removed this paragraph. We agree that the information provided in it on the “current dependent” block proposal of Bucchi et al 2002, 2013 has also been given, to a sufficiently detailed degree, in the text explaining Fig 10 (to which a quotation of Bucchi et al 2013 has been added).

10. It may also be important that even though HCN4 is the main isoform (gene) in the SA node, there are key differences in the function of the funny current of the SA node and currents produced by overexpression of the HCN4 channel. For example, I think the funny current produced in heterologous cells that express HCN4 open and close more slowly than the funny current of sinoatrial pacemaker cells? An interesting idea that HCN1 may be expressed in the node and contribute to those differences has been proposed by the DiFrancesco group.

We have added a sentence mentioning the evidence for expression of different isoforms (HCN1 and HCN2) in the SAN of various species and as a consequence the likely contribution of heterometric constructs to the funny channel properties.

11. This section seems like it is hanging as much of the functional experiments are discussed in the later section on structure. Is the purpose of this section to introduce the cloning of the four isoforms? Overall, I am not clear on the purpose of this section?

As explained above at point 7, this sub-section is now moved ahead below the heading “Structure mirroring function”.

12. For cloning HCN, I noted that papers by Santoro et al and Ludwig et al are cited here. However, there is another paper on HCN that was published back-to-back with the Ludwig paper in the same issue of Nature by Gauss et al, which you could also cite.

Quotation of Gauss et al 1988 added.

13. Structure mirroring function

This is the main section that connects original studies with the recent work on structures.

Page 11. "Does the new molecular insight validate or disprove the features described over four decades ago? " I am not sure that the early experimental findings would be invalidated or disproved by more recent work? I think they inform each other?

Some of the funny current features, for example the dependence of channel conductance on external K, interpreted in 1982 according to a Michaelis-Menten activation model with an equilibrium constant of about 44 mM, or the voltage dependence of Cs block, interpreted in 1982 according to a block model where Cs ions cross a specific fraction of the voltage gradient before reaching its binding site (about 0.71), can indeed find confirmation or disproof in structural data. Similarly, the interpretation of the current-dependence of ivabradine block as due to electrostatic interaction between the tertiary ammonium ion of ivabradine and a K⁺ ion when coordinated at the lowermost coordination site of the permeability filter can also be either confirmed or disproved.

We agree anyway that the sentence can be misleading and have modified it to stress that the validation or disproof concerns more the interpretation of data, rather than the data themselves.

14. "On page 11 As is described below, the molecular properties of HCN4 channels faithfully confirm original findings." As I mention above, I think that some of the interesting observations of the original studies are not readily explained by the cloning, functional expression and structure?

See above the answer to point 1.

15. So, I am not sure that they faithfully confirm original studies. I think that this also suggests that the structure confirms function whereas I think this works both ways?

Concerning the fact that structure and function work both ways, it is important to stress that structure by itself is not sufficient to provide evidence to confirm or disprove original studies. Indeed, as we have highlighted in this review, the structures represent key milestones for the use of cutting-edge in-silico methodologies, such as MD simulations, that can be eventually used to validate previous functional results.

16. In the second paragraph of page 11, "More recently, the cryo-EM technique has allowed to solve high-resolution structures of the entire HCN1 (Lee and MacKinnon, 2017, 2019; Burtscher et al., 2024). "

I am not sure the entire structure was used in either study? My understanding is that there were pieces removed from the HCN1 channel perhaps to improve expression? Is this also true for HCN4 and HCN3?

We considered the cryo-EM structures of HCN1, HCN3 and HCN4 as "entire structures" because they derived from proteins only missing an unstructured C-terminal sequence, which being unfolded, could not have been anyhow resolved. Indeed, in all cryo-EM structures of HCN1, 3 and 4 the unfolded N and C termini are not visible.

17. In the structure of HCN4 solved by cryo-EM, an open channel is also thought to result as well as a closed channel. Presumably, the cross-membrane voltage is 0 mV in the cryo-EM structure. I think this may require some explaining as there is also functional data suggesting the channels may remain open at depolarized voltages? It also raises the question of what happens to the pore when the voltage sensors move upon hyperpolarization? Does it just further facilitate opening or stabilize the open conformation?

We thank the reviewer for these comments. They prompted us to better describe the models of HCN1 (Lee and MacKinnon, 2019) and HCN4 (Saponaro et al., 2021a) used in this review to analyze atomistic details of voltage sensor architecture and movements underlying channel opening.

The reviewer is right, both structures were solved in the absence of voltage, a condition that does not allow voltage-dependent opening of HCN channels. Nonetheless, such movements were chemically induced, either by forcing S4 of HCN1 into a hyperpolarization-activated like position through chemical linkage (Lee and MacKinnon, 2019), or, in the case of HCN4, via the membrane mimetic amphipols, which chemically destabilize S4 - S5 interaction (Saponaro et al., 2021a).

We have now better clarified that, while in the HCN1 structure the S4 downward movements does not induce pore opening as expected (Lee and MacKinnon, 2019), in the HCN4 structure a similar downward sliding of the voltage sensor does cause pore opening (Saponaro et al., 2021a).

Regarding whether voltage sensor movements facilitate opening or stabilize the open, the currently accredited model for all voltage-dependent channels is that voltage sensor movement actively determines pore opening. In this light, since pore opening events of funny channels are hindered at depolarized potentials, while occurring upon hyperpolarization (DiFrancesco, 1986), this implies that in HCN the hyperpolarization-dependent movement of the voltage sensor allows pore opening. In general, whether such “permission” by the voltage sensor consists of the pore being left free to open or being forced to open, this is still debated and cannot be clarified through the 3D structures, which represent “fix images” of the protein.

18. On page 14,

"The atomic model of HCN1 and, in particular, the open pore of HCN4, have solved this apparent contradiction by showing that the difference in the permeation properties between HCN and K⁺ channels does not depend on the primary sequence, but rather on the architecture and, most importantly, on the dynamics of their respective SFs."

This is a bit confusing because, ultimately, I think the architecture and dynamics do depend on the primary sequence. So, I am not quite sure what this phrase is intending?

We have now clarified that the permeation properties of HCN and K⁺ channels depend essentially on the way the dynamics of the SF is affected by the surrounding residues, so they may differ even if the primary sequence is the same.

19. Page 19. "Thus, it is to be expected that the molecular mechanism of HCN channel gating requires, at a given location, a short-range physical interaction allowing the crosstalk of the two sensor domains CNBD and VSD."

I think this is an interesting finding but I am not sure it is expected because both the voltage sensing regions and CNBD are attached to the pore? What does a more direct connection achieve than an indirect one does not?

In order to be correctly interpreted, the paragraph quoted by the reviewer needs to be linked to the one just above it: "These data imply a coupling between two distant domains, i.e., the cytosolic, C-terminal CNBD and the transmembrane voltage sensor domain (VSD)". Indeed, given the several functional indications that the voltage sensor influences the affinity of cAMP binding to the CNBD bypassing the pore (Hummert et al., 2018; Kusch et al., 2010; Wu et al., 2011), as we have written, "it is to be expected that the molecular mechanism of HCN channel gating requires, at a given location, a short-range physical interaction allowing the crosstalk of the two sensor domains CNBD and VSD". As we have described, thanks to the cryo-EM structures of HCN1 and HCN4 this postulated connection that bypasses the pore domain was recently identified (see Fig. 9) and also biochemically and functionally confirmed (Porro et al., 2019; Saponaro et al., 2021a).

20. On page 20. It is mentioned that the HCN1 isoform is insensitive to cAMP, which may not be clear. I think the maximum effect of cAMP is smaller than for the HCN2 and HCN4 isoform. However, there is data that suggests the potency of cAMP is similar between HCN1 and HCN2 (I think in reference 94).

The reviewer is right. To correctly follow Wainger et al., 2001 and Altomare et al., 2003, we have now rephrased the sentence stating that in HCN1 the degree of cAMP potentiation is smaller than in HCN2 and HCN4.

21. Page 10. Two papers are cited that reported the first cloning of HCN subunits.

Quotation to Gauss et al 1998 now added.

22. On page 11, sensitivity to cAMP is again used. Perhaps, it needs to be defined?

As above, we have rephrased considering the differences in the extent of cAMP modulation among isoatypes.

23. Page 17 "Both these electrophysiological findings are now supported by MDs". MD simulations?

Corrected.

24. Page 19. "All experimental data could be faultlessly fitted" perhaps well-fitted?

Reworded with "correctly fitted".

25. Page 20 "a newly defined cytosolic "HCN domain" (HCND, Fig. 9B), at the channel N-terminus (Lee and MacKinnon, 2017)." This domain was actually noted a few years ago but its structure was solved in 2017.

The reviewer is right. We added a sentence and the reference Tran et al (2002).

Dear Professor DiFrancesco,

Re: JP-TR-2025-287209R1 "Structure mirroring function: What's the "matter" with the funny current?" by Dario DiFrancesco and Andrea Saponaro

We are pleased to tell you that your paper has been accepted for publication in The Journal of Physiology.

*****IMPORTANT*****

Your article has been identified as being suitable for our forthcoming special issue, 'Pacemaking in multi-cellular organ systems'. The Reviewing Editor would very much like to include your paper in this special issue.

Would you be happy with that?

We hope to finalise the special issue in the next few weeks. Your article would, however, be published online as soon as it has been through the production process.

Could you let us know, please?

Please email your reply to us at: jp@physoc.org

Authors should note that it is too late at this point to offer corrections prior to proofing. Major corrections at proof stage, such as changes to figures, will be referred to the Editors for approval before they can be incorporated. Only minor changes, such as to style and consistency, should be made at proof stage. Changes that need to be made after proof stage will usually require a formal correction notice.

Yours sincerely,

Laura Bennet
Senior Editor
The Journal of Physiology

P.S. - You can help your research get the attention it deserves! Check out Wiley's free Promotion Guide for best-practice recommendations for promoting your work at www.wileyauthors.com/eoo/guide. You can learn more about Wiley Editing Services which offers professional video, design, and writing services to create shareable video abstracts, infographics, conference posters, lay summaries, and research news stories for your research at www.wileyauthors.com/eoo/promotion.

IMPORTANT NOTICE ABOUT OPEN ACCESS: To assist authors whose funding agencies mandate public access to published research findings sooner than 12 months after publication, The Journal of Physiology allows authors to pay an Open Access (OA) fee to have their papers made freely available immediately on publication.

You can check if your funder or institution has a Wiley Open Access Account here: <https://authorservices.wiley.com/author-resources/Journal-Authors/licensing-and-open-access/open-access/author-compliance-tool.html>.

EDITOR COMMENTS

Reviewing Editor:

Ready for full acceptance.

REFEREE COMMENTS

Referee #1:

No further questions.

Referee #2:

Thank you for your clear and thoughtful responses to my previous comments and questions.